# INSTRUCTX: TOWARDS UNIFIED VISUAL EDITING WITH MLLM GUIDANCE

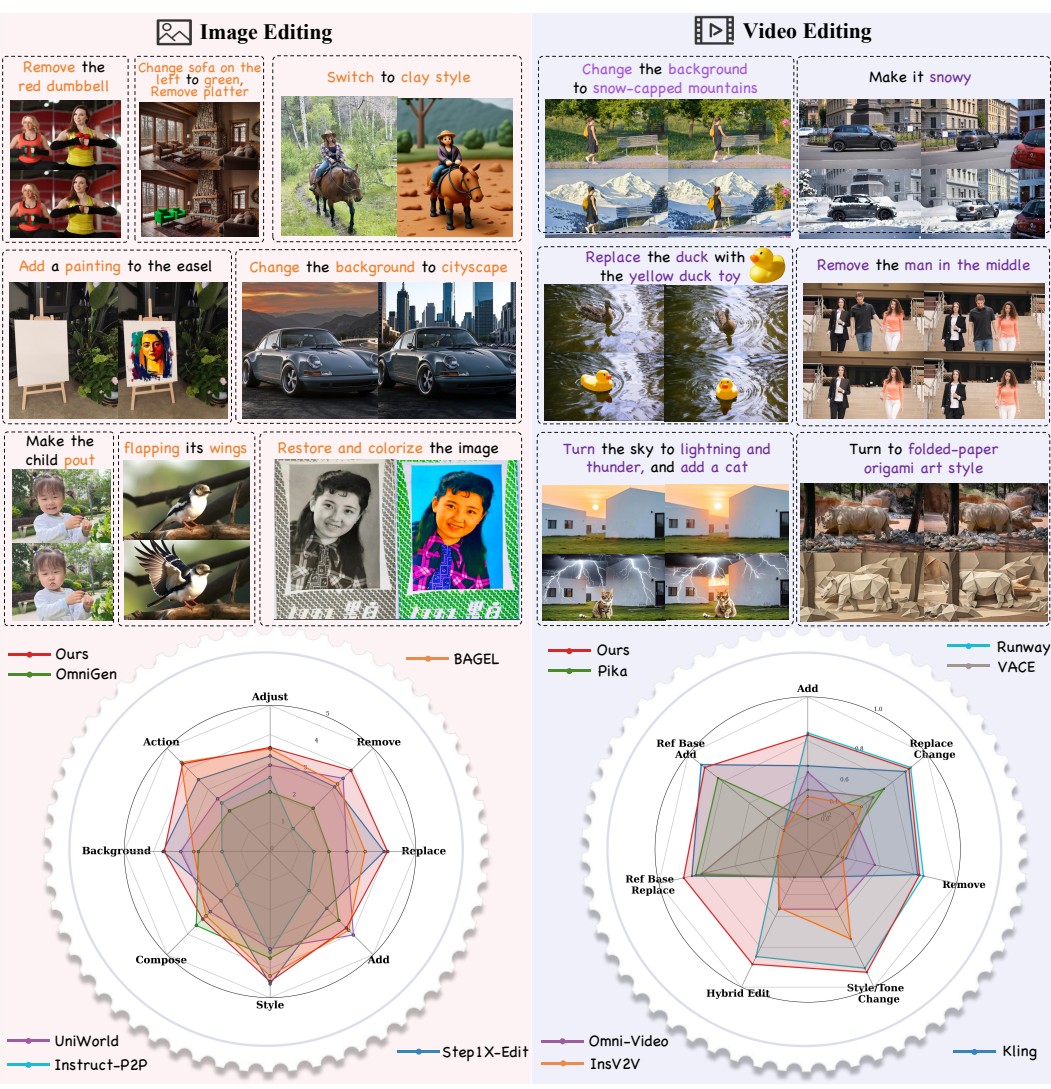

Figure 1: Showcase of InstructX. The bottom panel presents state-of-the-art performance of InstructX in image and video editing.

## ABSTRACT

With recent advances in Multimodal Large Language Models (MLLM) showing strong visual understanding and reasoning, interest is growing in using them to improve the editing performance of diffusion models. Despite rapid progress, most studies lack an in-depth analysis of MLLM design choice. Moreover, the integration of MLLM and diffusion models remains an open challenge in some difficult tasks, *e.g.*, video editing. In this paper, we present InstructX, a unified framework for image and video editing. Specifically, we conduct a comprehensive study on integrating MLLM and diffusion model for instruction-driven editing across di-

verse tasks. Building on this study, we analyze the cooperation and distinction between images and videos in unified modeling. *(1)* We show that training on image data can emerge video editing capabilities without explicit supervision, thereby alleviating the constraints imposed by scarce video training data. *(2)* By incorporating modality-specific MLLM features, our approach effectively unifies image and video editing tasks within a single model. Extensive experiments demonstrate that our method can handle a broad range of image and video editing tasks and achieve state-of-the-art performance.

# 1 INTRODUCTION

Recent research demonstrates a growing trend toward developing unified models that integrate multimodal understanding with generation. For example, systems for text-to-image generation Xie et al. (2024); Zhou et al. (2024); Chen et al. (2025a), image editing Deng et al. (2025); Lin et al. (2025); Liu et al. (2025); Wu et al. (2025) and video editing Liang et al. (2025); Wang et al. (2024a); Yu et al. (2025), have achieved impressive results. However, how to effectively integrate Multimodal Large Language Models (MLLM) with diffusion models, thereby leveraging their understanding and reasoning capabilities to aid visual editing tasks, remains an open question.

Typical integration paradigms include: (1) autoregressive visual generation Chen et al. (2025b); Lu et al. (2023); Qu et al. (2025) with discrete visual tokenizers Lee et al. (2022); Yu et al. (2021), (2) hybrid AR–diffusion approaches that unify an autoregressive loss for text and a diffusion loss for vision within a single transformer Zhou et al. (2024); Ma et al. (2025b); Shi et al. (2024a); Deng et al. (2025), and (3) using an MLLM backbone combined with an external diffusion model as the visual decoder Dong et al.; Ge et al. (2024); Sun et al. (2024); Pan et al. (2025). In this paper, we adopt an external diffusion model framework because it typically converges quickly, requires minimal changes, and delivers competitive performance. Although several visual editing works have been developed under this paradigm Lin et al. (2025); Wu et al. (2025); Liu et al. (2025); Yu et al. (2025), the role of MLLMs in the editing pipeline has yet to be examined in sufficient detail. Recently, MetaQuery Pan et al. (2025) introduces a set of learnable queries that act as an interface between MLLM and diffusion models. Moreover, MetaQuery employs a large connector (1.6B parameters) between the MLLM and the diffusion model while keeping the MLLM parameters fixed. However, a consensus has not been reached on the optimal integration of MLLM with diffusion models for editing tasks. Specifically, debates persist regarding several key design choices: whether to directly utilize all last hidden states or compress them into meta-query features; whether the connector should be a large transformer or if a small Multi-Layer Perceptron (MLP) suffices; and whether the MLLM itself requires fine-tuning. In this paper, we conduct a comprehensive study and validate a central hypothesis: to fully leverage the understanding capabilities of MLLMs, they should not be treated merely as feature extractors; instead, editing should be primarily realized within the MLLM, rather than delegated to a subsequent large connector.

Collecting high-quality video data remains a bottleneck for video editing. Early works Qi et al. (2023); Cong et al. (2023); Wu et al. (2023) perform video editing through zero-shot strategies, but they are often limited in generation quality and generalizability. Other methods Ku et al. (2024); Ouyang et al. (2024); Mou et al. (2024) transfer image editing capabilities to video by editing the first frame and propagating the changes, which is prone to content drift and loss. Recently, several methods Ye et al. (2025b); Zi et al. (2025b) have sought to construct video-editing datasets by training video-expert models; however, these approaches suffer from lengthy data-construction pipelines and low success rates. Noting that recent commercial models, such as GPT-4o OpenAI, have set a new standard for instructional image editing, we leverage large-scale, high-quality image editing data generated with these models to support video editing. This approach addresses both the scarcity of video-editing data and the narrow range of editing types. Specifically, we train on a mixture of image and video data and incorporate modality-specific MLLM features, unifying image and video editing within a single model. We observe that editing capabilities learned from image data transfer effectively to video editing without explicit supervision.

In summary, this paper has the following contributions:

- We present a unified framework that performs image and video editing within a single model. Our study analyzes the integration of MLLMs and diffusion models and offers insights for future research.

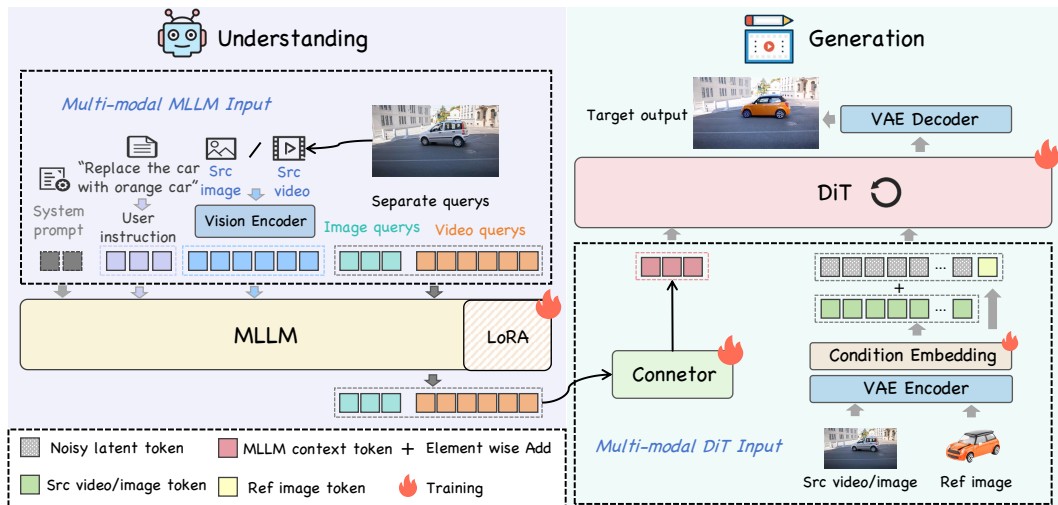

Figure 2: Overview of InstructX. The MLLM serves as the understanding module, generating editing guidance given the input instruction and visual inputs. The DiT serves as the generation module and connects to the MLLM via learnable queries and an MLP connector.

- We discuss a simple yet effective approach to extend zero-shot video editing capabilities via image training data. This design allows our method to tackle a wider range of tasks than existing open-source or closed-source methods.

- Extensive experiments show that our method achieves state-of-the-art performance across diverse image and video editing tasks.

## 2 RELATED WORK

### 2.1 INSTRUCTIONAL IMAGE AND VIDEO EDITING

Text-guided image editing significantly improves the convenience of visual manipulation by enabling users to modify images through natural language commands. Earlier approaches Nam et al. (2018); Li et al. (2020); Fu et al. (2020) primarily rely on GAN frameworks Goodfellow et al. (2020), often being constrained by limited realism and narrow domain applicability. The advent of diffusion models Ho et al. (2020) enables high-quality image editing via text. Early works learn from synthetic input-goal-instruction triples Brooks et al. (2023) and with additional human feedback Zhang et al. (2024b) to follow editing instructions. Fu et al. (2023) investigates how MLLM facilitate edit instructions. Recently, as MLLM grows in scale and demonstrates stronger capabilities in instruction understanding, several unified modeling approaches Lin et al. (2025); Liu et al. (2025); OpenAI; Zeng et al. (2025) are proposed, improving the performance of image editing.

When it comes to video editing, the challenge becomes significantly harder. Limited by model capabilities and training data, early research Qi et al. (2023); Cong et al. (2023); Wu et al. (2023) primarily relies on zero-shot or one-shot approaches based on image diffusion models. Later, with the performance scale-up of video diffusion models, several downstream tasks emerge, leveraging pre-trained video diffusion models. Examples include video inpainting Zi et al. (2025c); Bian et al. (2025), video try-on Fang et al. (2024); Zuo et al. (2025), and video addition Tu et al. (2025); Zhuang et al. (2025). Recently, some unified modeling methods Liang et al. (2025); Yu et al. (2025); Ye et al. (2025b) are proposed for video editing. However, these methods are constrained by manual priors, such as specifying editing areas and motion trajectories. Instruction-based editing offers a more convenient way. Early research, InsV2V Cheng et al. (2023), adapt image instruction editing model Brooks et al. (2023) to generate video training pairs. However, due to limitations in data quality, the editing results are often unsatisfactory. Very recent studies Tan et al. (2025) integrate the comprehension capabilities of MLLM into video editing. However, model designs are often not justified experimentally or very briefly, and the scope of tasks remains limited by the training data.

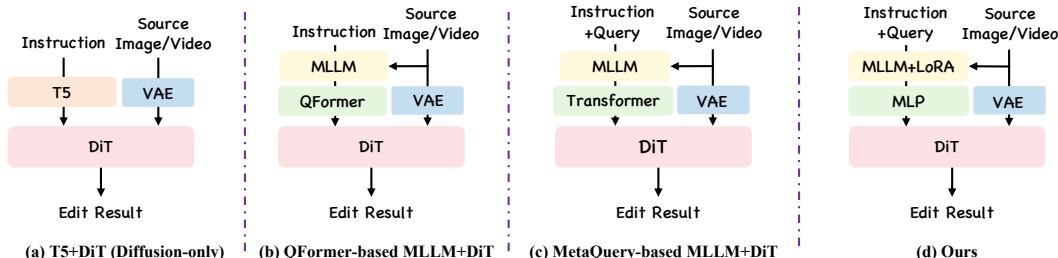

Figure 3: Different design choices for unified editing modeling.

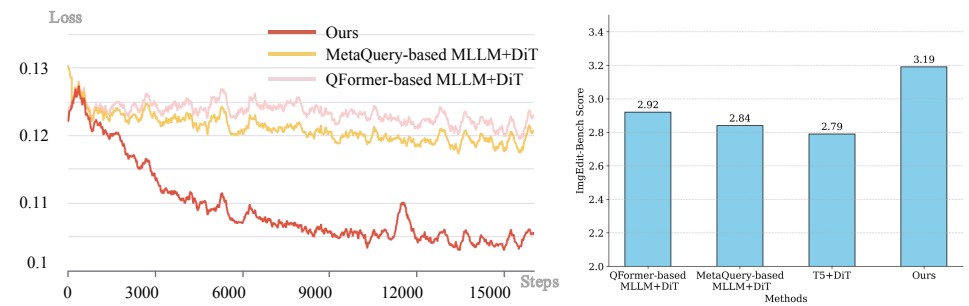

Figure 4: Illustration of alignment ability (left) and editing performance (right) for different design choices.

## 2.2 UNIFIED UNDERSTANDING AND GENERATION MODELS

Recently, extensive attempts extend the success of multimodal understanding to multimodal generation. Some works learn to regress image features Ge et al. (2024); Sun et al. (2023); Tong et al. (2024); some works auto-regressively predict the next visual tokens Jin et al. (2023); Team (2024); Xie et al. (2024); and some works Zhou et al. (2024); Ma et al. (2025b); Shi et al. (2024a); Deng et al. (2025) employ diffusion objective for visual generation and autoregressive objective for text generation. In this field, using a connector Dong et al.; Ge et al. (2024); Sun et al. (2024) to bridge the understanding model and diffusion model is a strategy for rapid convergence, while also delivering promising results. Recent work on MetaQuery Pan et al. (2025) introduces a useful bridging method through a set of learnable queries. However, for visual editing, several questions arise: whether to use all final hidden states directly or compress them into meta-queries; whether a large connector is necessary; and whether freezing the MLLM is sufficient. We study these questions in this work.

## 3 METHOD

### 3.1 OVERVIEW

An overview of InstructX is presented in Fig. 2. Recall that our goal is to build a unified architecture for image and video editing by leveraging the comprehension capabilities of MLLM. To this end, we employ a multimodal understanding model, *i.e.*, QWen2.5-VL-3B Bai et al. (2025), to embed the editing instruction and source image/video. Inspired by MetaQuery Pan et al. (2025), we append a set of learnable queries to the MLLM input sequence to extract editing information and retain only the meta-query features from the MLLM output. Wan2.1-14B Wan et al. (2025) is used as the decoder for the edited output. The produced queries from the MLLM are fed into a two-layer MLP connector, and are subsequently used to replace the text embeddings in the DiT model. To enhance the consistency between the edited result and the source image/video, we add the VAE encoding of the original image/video to the noisy latent. For tasks involving a reference image, we concatenate the VAE features of the reference image to the noisy latent along the sequence dimension.

### 3.2 ARCHITECTURE CHOICE

**Different choices**. As noted above, integrating understanding and generation models exposes many design choices that are often not empirically justified in prior work. We conduct a comprehensive study of these structural design choices. In Fig. 3, we compare several instruction-editing architec-

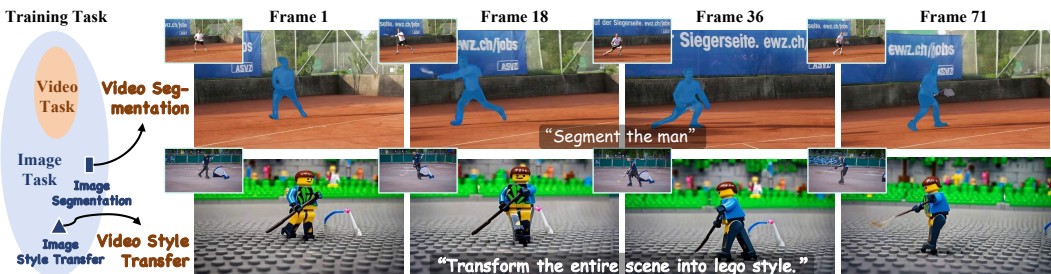

Figure 5: Illustration of three training stages of our methods.

Figure 6: Examples for emergent video editing capabilities through image data.

tures: (a) Instructions are encoded by the native T5 text encoder Chung et al. (2024) and fed directly into the diffusion model, *i.e.*, diffusion-only setting. (b) The last hidden states of the MLLM are encoded by QFormer Li et al. (2023) into fixed-length representation (*i.e.*, 256 tokens), which is then input to DiT. (c) The MetaQuery Pan et al. (2025) structure uses learnable queries to extract editing information from the MLLM and employs a large connector to bridge the MLLM and the DiT. (d) The architecture adopted in this work. It uses the same learnable queries as MetaQuery, fine-tunes the MLLM LoRA, and employs a simple two-layer MLP as the connector between MLLM and DiT. **Comparsion**. We validate the performance of different structure choices from two aspects. (1) Feature alignment capability. Due to the gap between the MLLM text space and the diffusion generation space, previous works Dong et al.; Ge et al. (2024) usually incorporate a pre-training stage to align these two spaces. Here, we freeze the DiT and train different designs on image editing task. The left part of Fig 4 shows that solely relying on a large-scale connector or a learnable query mechanism for the understanding-generation alignment converges slowly. Partially involving MLLM in feature alignment via LoRA Hu et al. (2022) accelerates convergence. Note that the T5 features are already aligned with DiT, hence not involved in this stage. Upon completion of the alignment stage, we unfreeze the DiT for continued training and evaluate the performance of various methods on ImgEdit-Bench Ye et al. (2025a). The right part in Fig. 4 also shows an advantage of the design choice in this paper. We also provide a further discussion on the gains of MLLM in the appendix A.8. **Other details**. Moreover, to model images and videos in a unified architecture while distinguishing between the two modalities, we introduce separate sets of learnable queries for each: 256 queries for image inputs and 512 queries for video inputs. Note that for video input, we specifically sample 13 frames to serve as input to the MLLM. Further experimental details are provided in Sec. 4.4.

## 3.3 TRAINING STRATEGIES

**Three stages**. As shown in Fig. 5, the training process is divided into three stages: feature alignment training, full-data training, and quality fine-tuning. **Stage 1:** The target of the first stage is to align the feature space of the MLLM with the generation space of the DiT. During this stage, we only train the learnable query, the LoRA in the MLLM, and the MLP connector on the image-instruction data. After this stage, the model acquires a rough instruction-based editing capability. However, due to the coarse-grained visual information in the MLLM, the editing results exhibit poor consistency with the original image. **Stage 2:** The second stage has two objectives: (1) Improving the fidelity between

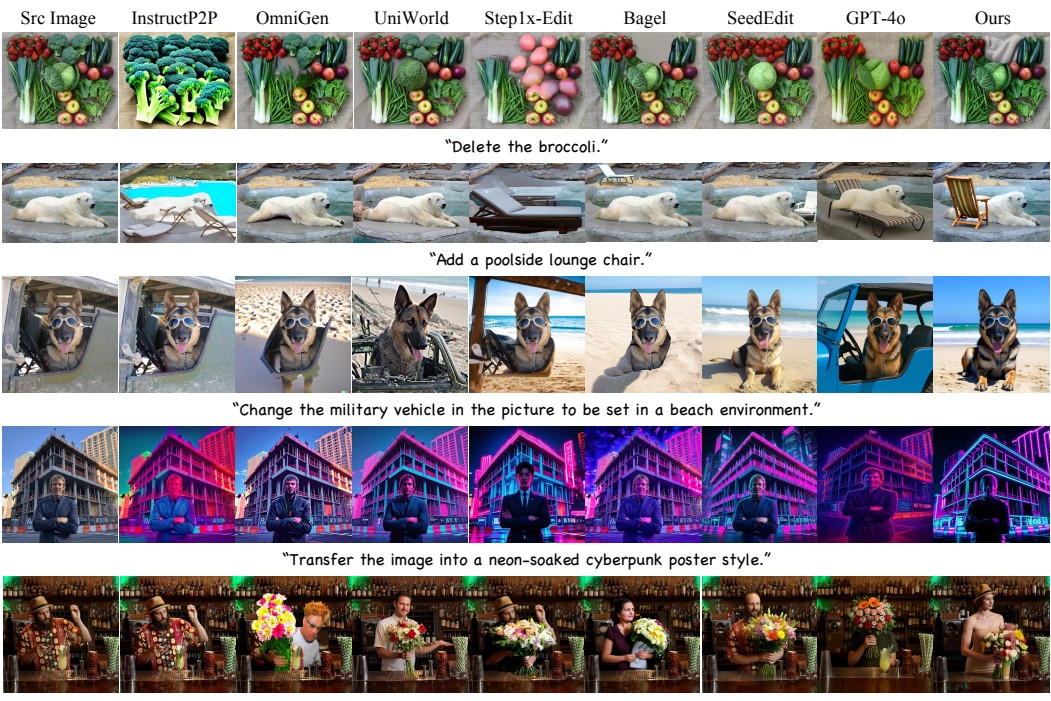

Figure 7: Visual comparsion between our InstructX and other methods on image editing tasks.

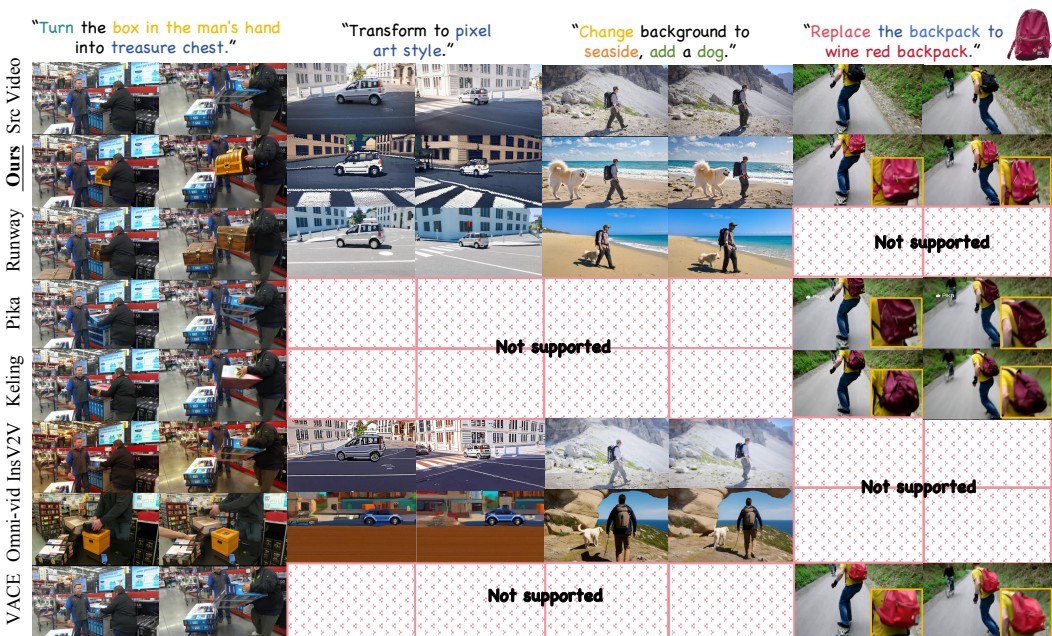

Figure 8: Visual comparsion between our InstructX and other methods on video editing tasks.

the editing results and the original visual input by incorporating VAE features, and (2) to enable the model to acquire unified and generalized image/video editing capabilities through full-data training. In this stage, we train the learnable query, the LoRA in the MLLM, the MLP connector, and the entire DiT. Note that mixing image and video training in this stage not only allows for unified modeling with a single model but also excites video editing capabilities that are difficult to obtain training data, by leveraging image data. As shown in Fig. 6, segmentation and style transfer tasks absent

from the video data but present in the image data. After mixed training, the model also acquires the capability for video style transfer. **Stage 3:** Although the model acquires unified image/video editing capabilities after the second stage, the generation quality is affected by some low-quality training data, resulting in the oily and plastic-like textures. To rectify this problem, we collect a small amount of high-quality training data and perform quality fine-tuning. As shown in the last row of Fig. 5, the generated results become more natural and aesthetically pleasing after quality fine-tuning. We use flow-matching Lipman et al. (2022) as the training objective in all stages.

**Training data**. For instruction-based image editing, we utilize large-scale open-source training data, including NHR-Edit Kuprashevich et al. (2025), X2Edit Ma et al. (2025a), and GPT-Image-Edit Wang et al. (2025b). For video editing, due to the lack of high-quality open-source video editing data, we develop a pipeline for synthesizing video-editing data. More details are provided in the appendix A.2.

## 4 EXPERIMENT

### 4.1 IMPLEMENTATION DETAILS

During training, we set the learning rate to $1 \times 10^{-5}$, with a global batch size of 128 for images and 32 for videos. In the first and second training stages, we iterate for $20,000$ steps each, while the third stage involves $5,000$ iterations. During the image/video mixed training, we sample video data with a probability of $0.6$ and image data with a probability of $0.4$.

### 4.2 EVALUATION DETAILS

For image editing, we compare different methods on two benchmarks: ImgEdit-Bench Ye et al. (2025a) and GEdit-Bench Liu et al. (2025). Specifically, on ImgEdit-Bench, we use GPT-4.1 OpenAI to score the editing results on a 1-5 scale. On GEdit-Bench, we employ Qwen2.5-VL-72B Bai et al. (2025) to evaluate the edited results across three metrics: instruction-following score (Q_SC), perceptual-quality score (Q_PQ), and overall score (Q_O). We compare our method with the well-known instruction-based image editing method (*i.e.*, InstructPix2Pix Brooks et al. (2023)), recent state-of-the-art approaches (*i.e.*, OmniGen Xiao et al. (2025), Uniworld Lin et al. (2025), Step1x-Edit Liu et al. (2025), Bagel Deng et al. (2025)), as well as several closed-source models (GPT-4o OpenAI, DouBao Shi et al. (2024b)).

For video editing, existing benchmarks(*e.g.*, UNICBench Ye et al. (2025b) and VACE-Benchmark Jiang et al. (2025)) primarily focus on target-prompt rather than instruction-prompt evaluation and provide few examples per task. To address the lack of instruction-based video-editing benchmarks, we introduce VIE-Bench, which comprises 140 high-quality instances across eight categories, covering both reference-free and reference-based edits. Further details are provided in Appendix Sec. A.1. Prior work commonly uses the CLIP text score to assess text–video alignment, which is effective for target-prompt settings but fails to capture instruction-following capability. Therefore, we adopt an MLLM-based judge using GPT-4o OpenAI to evaluate editing accuracy (instruction following), preservation (consistency with the source video), and quality (overall video quality). For reference-based editing, GPT-4o also assesses subject similarity to the reference image. All scores range from 1 to 10. The system prompts for the MLLM-based judge are provided in Appendix Sec. A.9. In addition, we employ VBench Zhang et al. (2024a) to evaluate video quality. We compare our method with the well-known baseline InsV2V Cheng et al. (2023), recent state-of-the-art approaches (VACE-14B Jiang et al. (2025), Omni-Video Tan et al. (2025)), and closed-source systems (Kling Keling (2025), Pika Pika (2025), Runway-Aleph Runway (2025)). For the removal task, we also evaluate against MiniMax-Remover Zi et al. (2025a) and DiffuEraser Li et al. (2025).

Table 1: **Comparison results on GEdit-Bench.** Q_SC, Q_PQ, and Q_O refer to the metrics evaluated by Qwen-2.5-VL-72B. The best and second best results are shown in **bold** and underlined respectively.

| Model | Community Model | Q_SC↑ | Q_PQ↑ | Q_O↑ |
|---|---|---|---|---|
| **Ours** | ✓ | **7.47** | 7.22 | 6.68 |
| Step1X-Edit | ✓ | 7.05 | 7.21 | 6.79 |
| Instruct-P2P | ✓ | 5.08 | 6.86 | 4.90 |
| OmniGen | ✓ | 6.33 | 6.96 | 6.04 |
| UniWorld | ✓ | 5.43 | **7.37** | 5.35 |
| Bagel | ✓ | 7.43 | 7.03 | **7.10** |
| SeedEdit 3.0 | ✗ | 7.92 | 7.39 | 7.57 |
| GPT-4o | ✗ | 7.98 | 7.73 | 7.83 |

Table 2: **Comparison results on ImgEdit-Bench.** "Overall" is calculated by averaging all scores across tasks. We use Qwen2.5-VL-72B for evaluation. The best and second best results are shown in **bold** and underlined respectively.

| Model | Community Model | Adjust | Remove | Replace | Add | Style | Compose | Background | Action | Overall↑ |
|---|---|---|---|---|---|---|---|---|---|---|
| **Ours** | ✓ | **3.56** | **3.92** | **4.03** | 3.7 | 4.45 | 3.27 | 3.63 | 4.24 | **3.85** |
| Step1X-Edit | ✓ | 3.27 | 3.13 | 3.91 | 2.75 | **4.53** | 2.38 | **3.67** | 3.48 | 3.39 |
| Instruct-P2P | ✓ | 2.53 | 1.11 | 1.50 | 1.89 | 3.44 | 1.61 | 1.65 | 2.35 | 2.01 |
| OmniGen | ✓ | 2.04 | 2.09 | 2.02 | 3.33 | 3.65 | **3.58** | 2.46 | 1.97 | 2.64 |
| UniWorld | ✓ | 2.95 | 3.54 | 2.64 | **4.04** | 3.33 | 2.91 | 3.07 | 2.55 | 3.13 |
| BAGEL | ✓ | 3.51 | 3.27 | 3.26 | 3.81 | 4.26 | 3.11 | 2.62 | **4.31** | 3.52 |
| SeedEdit 3.0 | ✗ | 2.43 | 4.27 | 4.33 | 4.40 | 4.51 | 4.32 | 3.58 | 4.62 | 4.06 |
| GPT-4o | ✗ | 4.15 | 4.54 | 4.49 | 4.84 | 4.63 | 4.30 | 4.87 | 4.22 | 4.51 |

Table 3: **Comparison results on VIE-Bench**. The best and second best results are shown in **bold** and underlined respectively.

| Task | | Method | | | VIE-Bench Score | | | | | Video Quality | |
|---|---|---|---|---|---|---|---|---|---|---|---|
| | | Model | Community Model | Instruction base | Instruct follow | Preser-vation | Quality | Similarity | Avg. | Smooth-ness | Aesthe-tics |
| **Video Edit** | | | | | | | | | | | |
| Add | | **Ours** | ✓ | ✓ | 8.446 | 8.683 | **7.919** | - | 8.349 | **0.991** | 0.558 |
| | | Kling | ✗ | ✓ | 6.000 | 8.230 | 5.576 | - | 6.602 | 0.988 | 0.519 |
| | | Runway | ✗ | ✓ | **8.607** | **8.913** | 7.823 | - | **8.447** | 0.990 | 0.557 |
| | | Omni-Video | ✓ | ✓ | 5.699 | 6.135 | 6.294 | - | 6.242 | 0.987 | **0.586** |
| | | InsV2V | ✓ | ✓ | 3.552 | 5.891 | 3.402 | - | 4.281 | 0.988 | 0.513 |
| | | VACE | ✓ | ✗ | 3.938 | 6.696 | 3.929 | - | 4.854 | 0.983 | 0.557 |
| Swap / Change | | **Ours** | ✓ | ✓ | 9.514 | **9.171** | 8.533 | - | 9.072 | 0.977 | **0.557** |
| | | Kling | ✗ | ✓ | 9.000 | 9.060 | 8.333 | - | 8.800 | **0.989** | 0.541 |
| | | Runway | ✗ | ✓ | **9.580** | 8.628 | **9.275** | - | **9.161** | 0.981 | 0.541 |
| | | Pika | ✗ | ✓ | 7.542 | 7.847 | 6.837 | - | 7.408 | 0.974 | 0.528 |
| | | Omni-Video | ✓ | ✓ | 4.733 | 4.856 | 4.656 | - | 4.748 | 0.981 | 0.556 |
| | | InsV2V | ✓ | ✓ | 5.304 | 6.428 | 4.971 | - | 5.567 | 0.977 | 0.530 |
| | | VACE | ✓ | ✗ | 6.171 | 7.552 | 6.199 | - | 6.640 | 0.976 | 0.534 |
| Remove | | **Ours** | ✓ | ✓ | 8.627 | 8.668 | 7.672 | - | 8.322 | 0.983 | 0.472 |
| | | Kling | ✗ | ✓ | 8.440 | 8.800 | 7.520 | - | 8.253 | **0.993** | 0.455 |
| | | Runway | ✗ | ✓ | **8.664** | **9.145** | **7.703** | - | **8.504** | 0.987 | 0.460 |
| | | Omni-Video | ✓ | ✓ | 6.004 | 5.970 | 4.807 | - | 5.593 | 0.989 | 0.417 |
| | | InsV2V | ✓ | ✓ | 1.209 | 3.769 | 1.322 | - | 2.098 | 0.982 | 0.517 |
| | | VACE | ✓ | ✗ | 1.812 | 3.877 | 2.359 | - | 2.682 | 0.983 | **0.535** |
| | | MiniMax | ✓ | ✗ | 6.963 | 7.518 | 6.037 | - | 6.839 | 0.985 | 0.467 |
| | | DiffuEraser | ✓ | ✗ | 6.346 | 6.807 | 5.576 | - | 6.243 | 0.986 | 0.465 |
| Style / Tone Change | | **Ours** | ✓ | ✓ | **9.650** | 9.099 | **8.839** | - | **9.196** | 0.972 | **0.560** |
| | | Runway | ✗ | ✓ | 9.583 | **9.200** | 8.616 | - | 9.133 | 0.982 | 0.547 |
| | | Omni-Video | ✓ | ✓ | 5.486 | 4.655 | 5.959 | - | 5.366 | **0.984** | 0.557 |
| | | InsV2V | ✓ | ✓ | 7.835 | 8.086 | 6.437 | - | 7.452 | 0.971 | 0.529 |
| Hybrid Edit | | **Ours** | ✓ | ✓ | **9.448** | **8.862** | **8.411** | - | **8.907** | 0.973 | **0.590** |
| | | Runway | ✗ | ✓ | 8.966 | 8.533 | 8.033 | - | 8.510 | **0.984** | 0.585 |
| | | Omni-Video | ✓ | ✓ | 5.444 | 5.066 | 5.766 | - | 5.425 | 0.978 | **0.608** |
| | | InsV2V | ✓ | ✓ | 5.033 | 5.966 | 4.966 | - | 5.321 | 0.975 | 0.541 |
| **Reference Base Video Edit** | | | | | | | | | | | |
| Ref Base Swap | | **Ours** | ✓ | ✓ | **9.210** | **9.201** | **8.221** | **9.190** | **8.955** | 0.978 | 0.549 |
| | | Kling | ✗ | ✓ | 8.830 | 8.910 | 8.120 | 8.510 | 8.592 | 0.988 | 0.522 |
| | | Pika | ✗ | ✓ | 8.438 | 8.665 | 7.656 | 8.447 | 8.301 | **0.989** | 0.462 |
| | | VACE | ✓ | ✗ | 8.312 | 8.542 | 7.442 | 7.654 | 7.987 | 0.976 | **0.550** |
| Ref Base Add | | **Ours** | ✓ | ✓ | 9.491 | 9.252 | 8.375 | **9.511** | 9.157 | 0.987 | **0.595** |
| | | Kling | ✗ | ✓ | **9.714** | **9.571** | **8.714** | 9.285 | **9.321** | **0.992** | 0.567 |
| | | Pika | ✗ | ✓ | 8.510 | 8.625 | 7.750 | 8.625 | 8.377 | 0.991 | 0.511 |
| | | VACE | ✓ | ✗ | 2.665 | 6.540 | 3.052 | 3.636 | 3.973 | 0.987 | 0.561 |

## 4.3 COMPARSION RESULT

Tab. 1 and Tab. 2 respectively present the comparsion results of our method and other methods on GEdit-Bench Liu et al. (2025) and ImgEdit-Bench Ye et al. (2025a). It can be observed that our method achieves competitive performance across multiple sub-tasks, and outperforms other open-source methods in terms of the overall score on ImgEdit-Bench. Fig. 7 demonstrates that in some

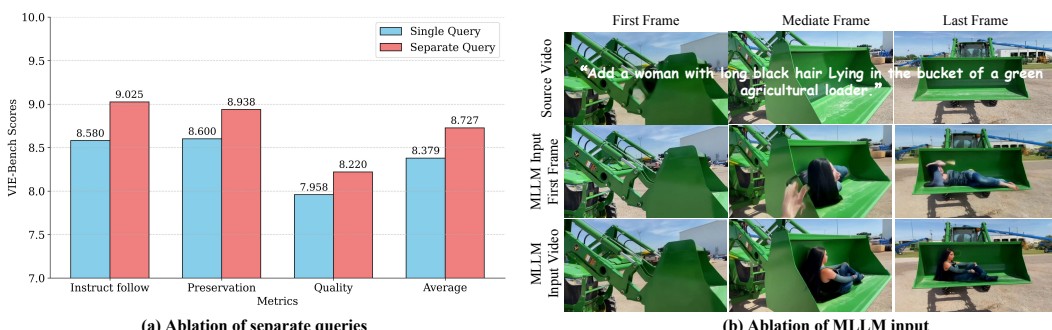

Figure 9: Ablation study of image/video independent queries (a) and MLLM inputs (b).

complex scenarios, such as removing broccoli from a cluttered pile of vegetables, methods like OmniGen Xiao et al. (2025), UniWorld Lin et al. (2025), and Step1x-Edit Liu et al. (2025) fail to recognize the target, while SeedEdit Shi et al. (2024b) and GPT-4o OpenAI produce editing results that lack consistency with the original image. Our method enables accurate removal while maintaining better consistency. Additionally, our advantages exist in cleaner background replacement and superior style consistency. We also conduct a user study in Sec. A.3 in appendix.

Table 3 shows that our method outperforms current open-source video-editing models on most metrics and remains competitive with state-of-the-art closed-source solutions. Specifically, our method attains the highest average scores on Style/Tone/Weather Change, Hybrid Edit, and Ref-Based Swap tasks among all methods, while scoring slightly below Runway Aleph on the Add, Swap/Change, and Remove tasks, and marginally below Kling on Ref-Based Add. Moreover, our method demonstrates leading advantages on several fine-grained evaluation dimensions. As shown in Fig. 8, on the fine-grained local editing task, our method achieves superior accuracy, while competing approaches either perform poorly on the handheld box replacement or fail to replace it. Our method also excels at style transfer and instruction following in hybrid edits. In reference-based editing, the backpack in our output shows higher similarity to the reference image. Additional visual comparisons are provided in Appendix Sec.A.10; we also report a user study in Appendix Sec. A.3.

### 4.4 ABLATION STUDY

We perform ablation studies on the design choice of unifying image and video editing: (1) whether to separate image and video queries; (2) whether the MLLM requires multi-frame video input. As shown in Fig 9 (a), the separate query setting achieves a higher score on VIE-Bench, as it better distinguishes the feature extraction for different modality information. Fig 9 (b) shows that if the MLLM only uses the first frame of the video to generate editing guidance, the editing results are prone to collapse in some complex scenarios, such as when the edited content appears in the middle of the video.

## 5 CONCLUSION

In this paper, we propose InstructX, a unified framework for image and video editing. Specifically, we conduct a comprehensive study on the design for the combination of MLLM and diffusion models, ultimately selecting the integration of Learnable Query, MLLM LoRA, and MLP Connector, which achieves faster convergence and superior performance. Furthermore, we explore mixed image-video training, which not only enables unified modeling for image and video editing but also expands the scope of video editing task. Additionally, we employ separate queries within the unified framework to better distinguish different modalities. We also introduce a MLLM-based video editing benchmark, *i.e.*, VIE-Bench, comprising 140 high-quality editing instances across eight categories. Extensive experiments demonstrate that our method outperforms the latest open-source image and video editing methods. Particularly, in video editing, InstructX achieves comparable performance to some closed-source editing methods while supporting a broader range of tasks.

**Limitation** Although InstructX demonstrates remarkable performance and appealing training efficiency, it is constrained by the pre-trained video DiT, making it difficult for high-resolution (e.g., >1080P) image/video editing. Although image data can excite zero-shot video editing capabilities, it is not a direct solution. However, it can serve as a temporary solution to address the current shortage of video data.

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

# A APPENDIX

## A.1 VIE-BENCHMARK DETAILS

As discussed in Sec. 4.2, given the scarcity of public video editing benchmarks, we build a high-quality, instruction-based video editing benchmark. Specifically,The source videos come from public datasets (*e.g.,* DAVIS Pont-Tuset et al. (2017), HumanVid Wang et al. (2024b)) and the web. All videos are 720P and 3–10 seconds long, covering indoor, outdoor, dynamic, animated, and portrait scenes. For each video, we used GPT-4o to generate 5 editing instructions, followed by careful manual curation to ensure that the instructions align with the original video content while retaining a degree of creativity. For reference-based editing tasks, the reference images are derived from the DreamBooth Ruiz et al. (2023) dataset. In total, our benchmark comprises eight fine-grained video-editing tasks with 140 editing examples. As shown in Tab. 4. The benchmark encompasses local video editing tasks—add, object swap, color change, and remove; global editing tasks—style change and tone/weather change; and reference base tasks-including reference base add and reference base swap.

## A.2 VIDEO SYNTHESIS PAIRED DATA PIPELINE

To construct high-quality paired training data for video editing, we develop a synthetic video-editing data pipeline covering the editing tasks: add, reference-based add, remove, swap, and reference-based swap. Source videos are drawn from Wang et al. (2025a). We use PySceneDetect to partition videos into single-scene clips, which serve as the original video. The data synthesis pipeline is shown in Fig. 10.

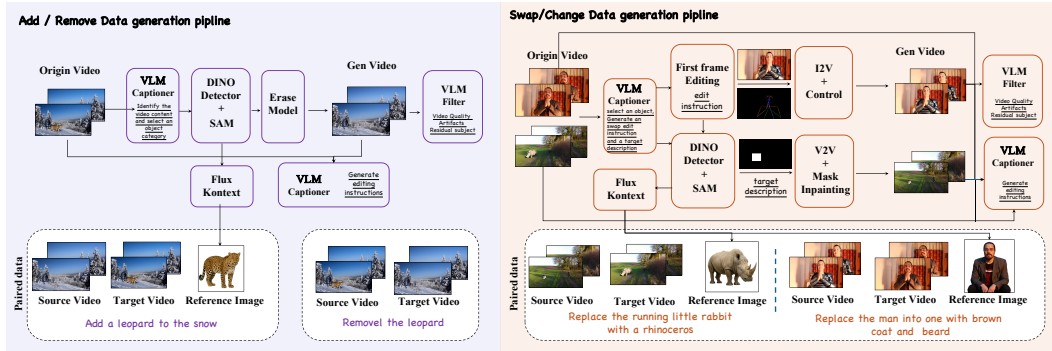

Figure 10: Pipeline for synthesizing paired video data.

Table 4: Editing Tasks in VIE-Bench.

| Edit Task | Sub Edit Task | Number |
|---|---|---|
| **Total** | | **140** |
| Local Edit | Object Swap | 25 |
| | Color Change | 10 |
| | Add | 30 |
| | Remove | 30 |
| Global Edit | Style Change | 10 |
| | Tone / Weather Change | 5 |
| Hybrid Edit | - | 10 |
| Reference Base Edit | Reference Base Swap | 10 |
| | Reference Base Add | 10 |

For the add and remove data. We first employ GPT-4o to analyze the video and identify a target subject category. Leveraging Grounding DINO Liu et al. (2023) and SAM Ravi et al. (2024), we

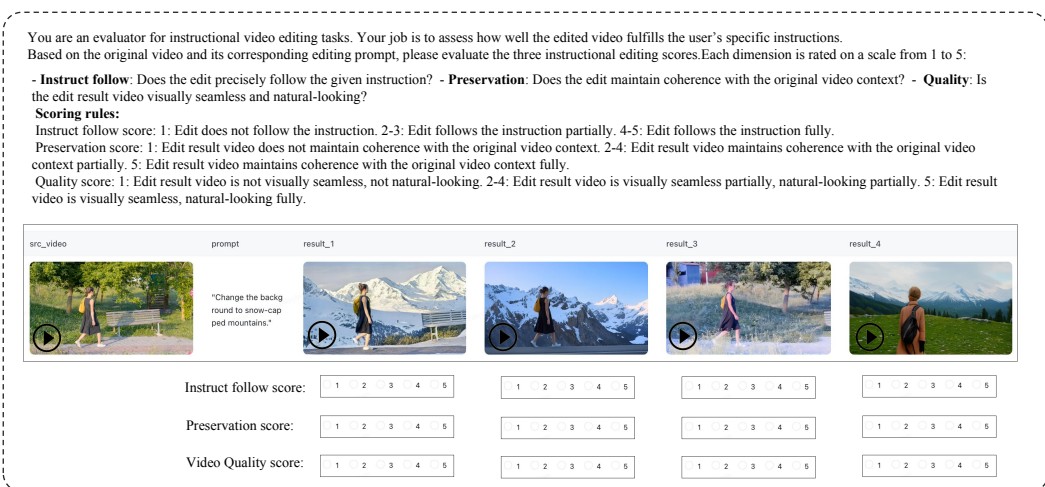

Figure 11: User study example.

segment the corresponding masks and then apply video erasure techniques Zi et al. (2025a) to remove the target subjects, with a MLLM-based filtering mechanism that avoids visual artifacts of inpainting. The original and erased videos are subsequently provided to GPT-4o.By swapping the roles of the original and generated videos, GPT-4o is prompted to produce "remove" and "add" editing instructions. We using Flux-Kontext Labs et al. (2025) to generate cross-pair reference images of the edit object, to form quadruples—source video, target video, reference image, and instruction prompt. Finally, the training set comprised 65K removal paired samples and 73K add paired samples.

For the swap and change data, we first apply an optical-flow-based analysis to partition videos into static-background and dynamic-background categories. Paired editing data are synthesized via two routes. First, we use GPT-4o to select a target subject category and to generate both the editing instruction and the target prompt. For human-centric, static-background videos, Flux-Kontext produces the edited first-frame image. Pose sequences of the characters are extracted with DW-pose Yang et al. (2023), after which a pose-driven image-to-video expert model generates a driven video used as the source video. The original video is treated as the target video, and these are provided to GPT-4o to obtain editing instructions. Additionally, we segment the target object in the first frame and use Flux-Kontext to generate cross-pair reference images of the edited target object, yielding paired training data composed of the source video, target video, reference image, and instruction prompt. For dynamic-background editing, a specially trained, mask-based video inpainting expert model is employed during video generation to construct editing triplets, ensuring consistency under substantial background changes and motion.We ultimately used 98K paired swap/change samples as training data.

### A.3 USER STUDY

We invited 30 professional image and video creators to serve as our user evaluation experts. For the image-editing tasks, we randomly selected 30 image-editing sample pairs from GEdit-Bench and 30 from ImgEdit-Bench, for a total of 60 pairs. For the video-editing tasks, we randomly selected 60 non-reference video-editing samples from VIE-bench. Our user study example is shown in Fig.11. Users rated 8 image-editing methods and 4 instruct-based video-editing methods on three dimensions, including 'Instruct follow', 'Preservation' and 'Quality'. All scores range from 1 to 5, and we averaged the ratings to obtain the final scores. The user study was carried out under blinded to reduce bias and promote fairness. Figs.12 and 13 indicate that our method outperforms current open source image and video editing methods in the user study and is competitive with the state-of-the-art closed source solution.

### A.4 MORE ABLATION STUDIES

**The impact of image data**. In Fig. 6 of the main paper, we demonstrate that the model can perform untrained video editing tasks under mixed image-video training. To further verify that this

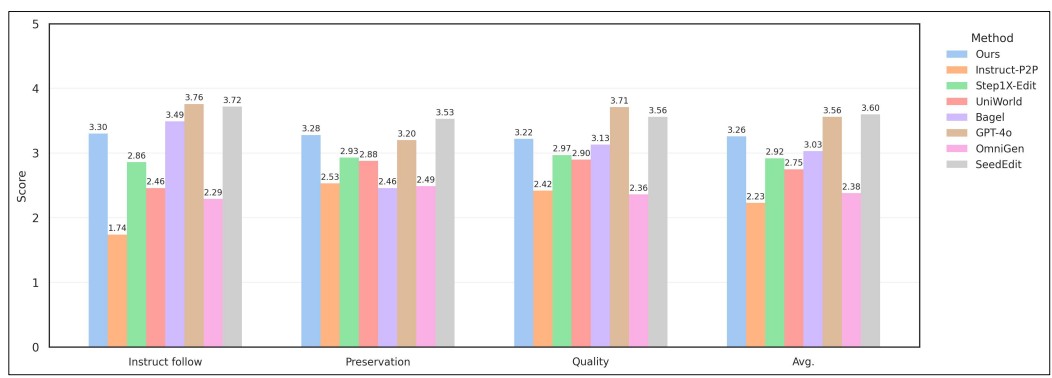

Figure 12: User study result of image edit.

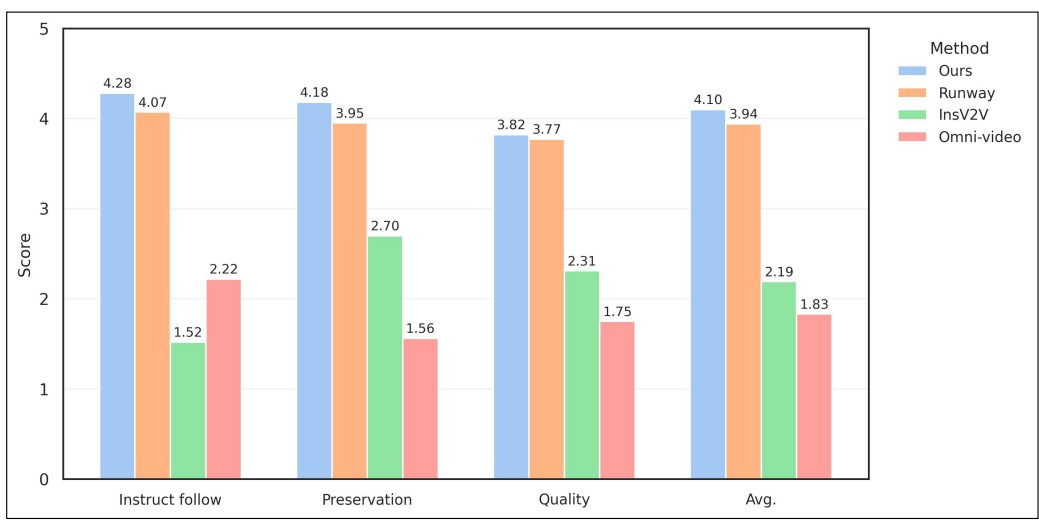

Figure 13: User study result of video edit.

improvement comes from the image data, Fig. 14 compares the editing results from training with both image and video data against those from training with video data alone. Note that the video data does not include training data for segmentation and style transfer. It can be seen that training without mixed image data fails to enable zero-shot video editing tasks. Furthermore, in Tab. 5, we present the impact of the image-video training mixture ratio on model performance, which shows insensitivity to the mixing ratio.

**The impact of video data**. The results in Fig. 15 demonstrate that using only image data disrupts the temporal consistency of video generation outcomes, leading to undesired flickering and artifacts. Moreover, the video editing performance obtained using only image training data are also unsatisfactory.

**The number of video queries**. In the main paper, Fig. 9(a) verifies the performance gain achieved by utilizing modality-independent queries. In this part, we further study the impact of the number of queries on performance. Specifically, we double the number of video queries. Tab. 6 indicates that an excessive number of queries does not yield a significant performance improvement, primarily because the VLM mainly provides high-level semantic information.

## A.5 LONG VIDEO EDITING PIPELINE

Our InstructX can also perform long video editing by modifying the inference pipeline. Specifically, we process long videos using a sliding window approach, where consecutive windows overlap at the tail frame of the previous window and the head frame of the next window. During editing, the editing result of the tail frame of the preceding window replaces the head frame of the subsequent

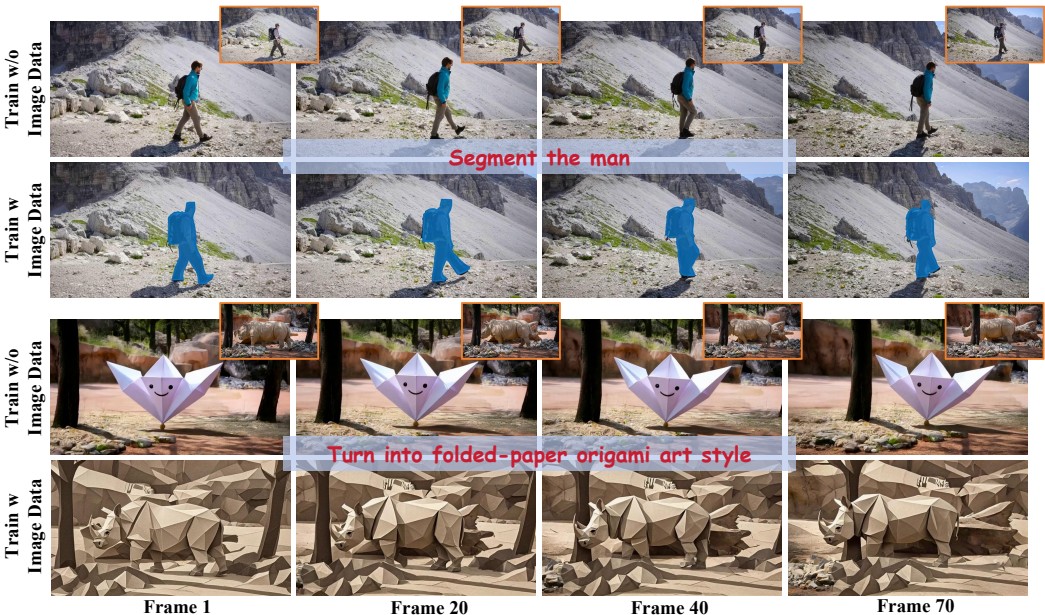

Figure 14: Comparison of zero-shot video editing capabilities with and without image training data. The first row demonstrates video segmentation, while the second row showcases video stylization.

Table 5: The impact of different image-video training mixture ratios on video editing performance.

| image:video | 5,000 iter | | | 10,000 iter | | |
|---|---|---|---|---|---|---|
| | Instruct follow | Preservation | Quality | Instruct follow | Preservation | Quality |
| 2:3 (paper setting) | 8.40 | 8.73 | 7.77 | 8.26 | 8.73 | 7.59 |
| 1:4 | 8.36 | 8.74 | 7.67 | 7.91 | 8.54 | 7.38 |
| 4:1 | 8.41 | 8.79 | 7.74 | 8.69 | 8.92 | 7.87 |

window to maintain consistency between windows. During testing, we use a 5-second window. Fig. 16 demonstrates the editing results for a 30-second, 30 FPS video. It can be observed that the transitions between windows are smooth. Therefore, our method can be extended to long video editing.

### A.6 HIGH-RESOLUTION VIDEO EDITING

Although we use 480P resolution data during training, we found that the model also has generalization capability for higher resolutions. Fig. 17 demonstrates the promising editing results at 1080P resolution.

### A.7 MORE DETAILS OF MODEL SIZE

In Tab. 7, we present the model size of representative image editing and video editing methods. It can be seen that the number of model parameters in our method is comparable to that of main-

Table 6: Ablation study of the number of video queries.

| | 5,000 iter | | | 10,000 iter | | |
|---|---|---|---|---|---|---|
| | Instruct follow | Preservation | Quality | Instruct follow | Preservation | Quality |
| 512 video query | 8.61 | 8.82 | 7.94 | 8.55 | 8.81 | 7.90 |
| 1024 video query | 8.85 | 8.98 | 8.10 | 8.70 | 8.91 | 8.02 |

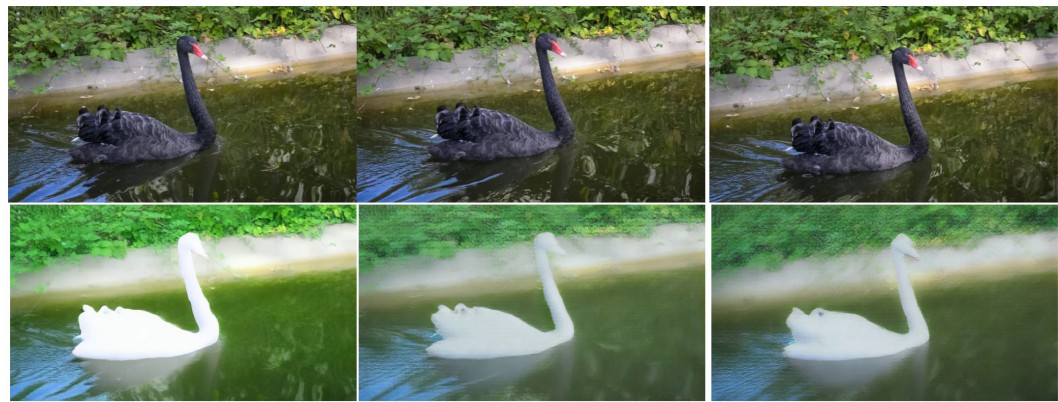

**Replace the blackswan with a white cat**

Figure 15: Video editing performance of InstructX trained solely on image editing data.

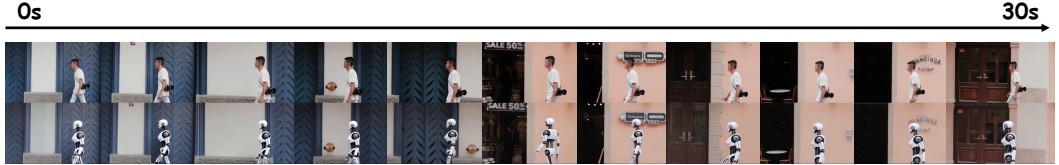

**Replace the man with a white robot**

Figure 16: The performance of InstructX in long video (30s) editing using a sliding window approach.

stream approaches. The comparaison in this paper demonstrate that our performance surpasses these methods.

Table 7: Model size of different methods

| Method | Model Size |
|---|---|
| OmniGen | 3.8B |
| Step1X-Edit | 12.5B |
| UniWorld | 12B |
| Bagel | 14.6B |
| Omni-Video | 11B |
| VACE | 14B |
| ours | 14B (DiT) + 0.6B (MLLM LoRA) |

### A.8 FURTHER DISCUSSION ON THE GAINS OF MLLM

In Fig. 18, we visualize the understanding ability gains of MLLM in visual editing. It can be observed that using only the diffusion model fails to comprehend some complex and tiny details, such as the books on the corner shelf and the plants in the corner. MLLM, however, can understand these elements quite well. In Fig. 19, we quantify the performance of using only diffusion for instruction-based editing versus MLLM+Diffusion across various tasks on ImgEdit-Bench Ye et al. (2025a). A noticeable gap can also be observed.

### A.9 MLLM-BASED JUDGE

We employ GPT-4o as MLLM-based judge. Figs.20 and 21 present the MLLM scoring prompts used in our paper for the video-editing and reference-based video-editing tasks respectively.

### A.10 MORE EXAMPLES

We show additional visual results in Figs. 22 - 27 .

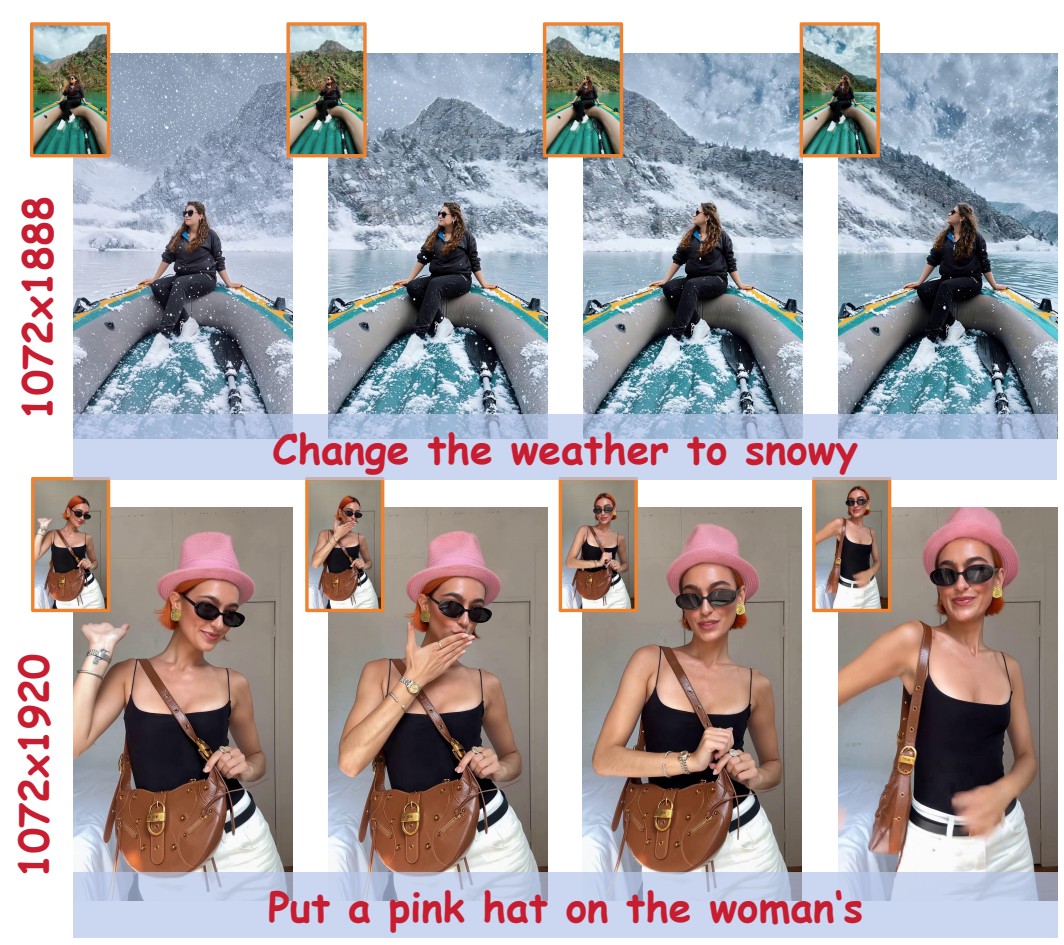

Figure 17: The editing performance of InstructX on 1080P videos.

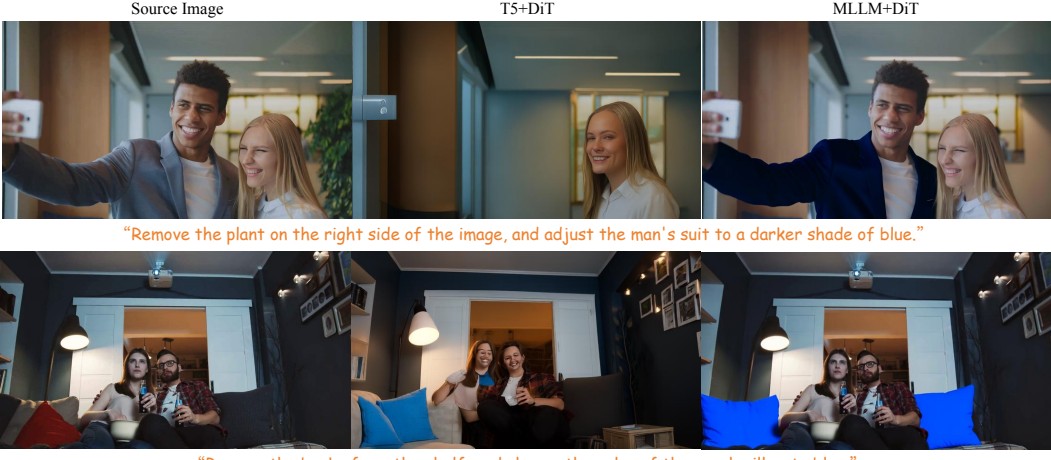

Figure 18: Comparison of understanding abilities between MLLM+Diffusion and Diffusion-only setting in instructional editing tasks.

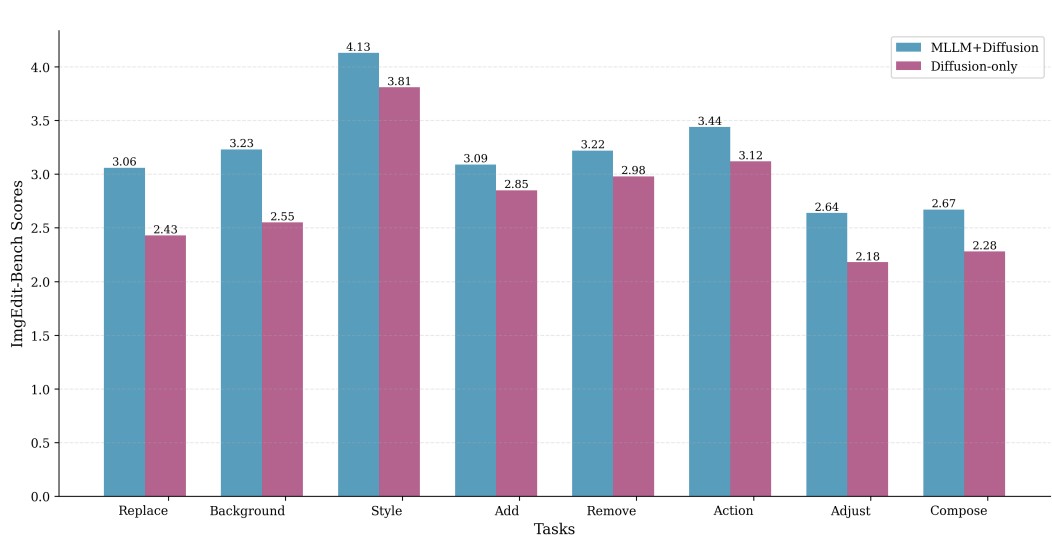

Figure 19: Comparison of understanding abilities between MLLM+Diffusion and Diffusion-only setting in instructional editing tasks.

```
"""
# **Role**
You are an evaluator for instructional video editing tasks. Your job is to assess how well the edited video fulfills the user's specific instructions.
# **Input**
1. The user's instruction
2. The original video (first video)
3. The edited video (second video)

# **Task**
 Please evaluate the instruct editing score:
 - **Instruct follow**: Does the edit precisely follow the given instruction? -   **Quality**: Is the edit result video visually seamless and natural-looking? - **Preservation**: Does the
 edit maintain coherence with the original video context?
 Scoring rules:
 Instruct follow score: 1-3: Edit does not follow the instruction. 4-6: Edit follows the instruction partially. 7-10: Edit follows the instruction fully.
 Quality score: 1-3: Edit result video is not visually seamless, not natural-looking and not aesthetics. 4-6: Edit result video is visually seamless partially, natural-looking
 partially, and aesthetics partially. 7-10: Edit result video is visually seamless fully, natural-looking fully, and aesthetics fully.
 Preservation score: 1-3: Edit result video does not maintain coherence with the original video context. 4-6: Edit result video maintains coherence with the original video
 context partially. 7-10: Edit result video maintains coherence with the original video context fully.
 Using the following Output format:

# **Output**
 Structure the output in JSON format with:
 - instruction: Repeat the user's instruction.
 - instruct follow score (1-10): Your score number
 - quality score (1-10): Your score number
 - preservation score (1-10): Your score number
 - reason: The reasons for the score you gave
"""
```

Figure 20: MLLM score system prompt for video edit.

```
"""
# **Role**
You are an evaluator for instructional video editing tasks. Your job is to assess how well the edited video fulfills the user's specific instructions.
# **Input**
1. The user's instruction
2. The reference image.
2. The original video (first video)
3. The edited video (second video)

# **Task**
Please evaluate the reference base instruct editing score: - **Instruct follow**: Does the edit precisely follow the given instruction? -   **Quality**: Is the edit result video
visually seamless and natural-looking? - **Preservation**: Does the edit maintain coherence with the original video context? - **Similarity**: The similarity between the editing
object in edited video(replace or add) and the reference image?
Scoring rules:
Instruct follow score: 1-3: Edit does not follow the instruction. 4-6: Edit follows the instruction partially. 7-10: Edit follows the instruction fully.
Quality score: 1-3: Edit result video is not visually seamless, not natural-looking and not aesthetics. 4-6: Edit result video is visually seamless partially, natural-looking
partially, and aesthetics partially. 7-10: Edit result video is visually seamless fully, natural-looking fully, and aesthetics fully.
Preservation score: 1-3: Edit result video does not maintain coherence with the original video context. 4-6: Edit result video maintains coherence with the original video
context partially. 7-10: Edit result video maintains coherence with the original video context fully.
Similarity score: 1-3: In the edited video (replaced or added), the similarity between the edited object and the reference image is low. 4-6: the similarity is medium . 7-10:
the similarity is high.
Using the following Output format:

# **Output**
Structure the output in JSON format with:
- instruction: Repeat the user's instruction.
- instruct follow score (1-10): Your score number
- quality score (1-10): Your score number
- preservation score (1-10): Your score number
- similarity score  (1-10): Your score number
- reason: The reasons for the score you gave
"""
```

Figure 21: MLLM score system prompt for reference base video edit.

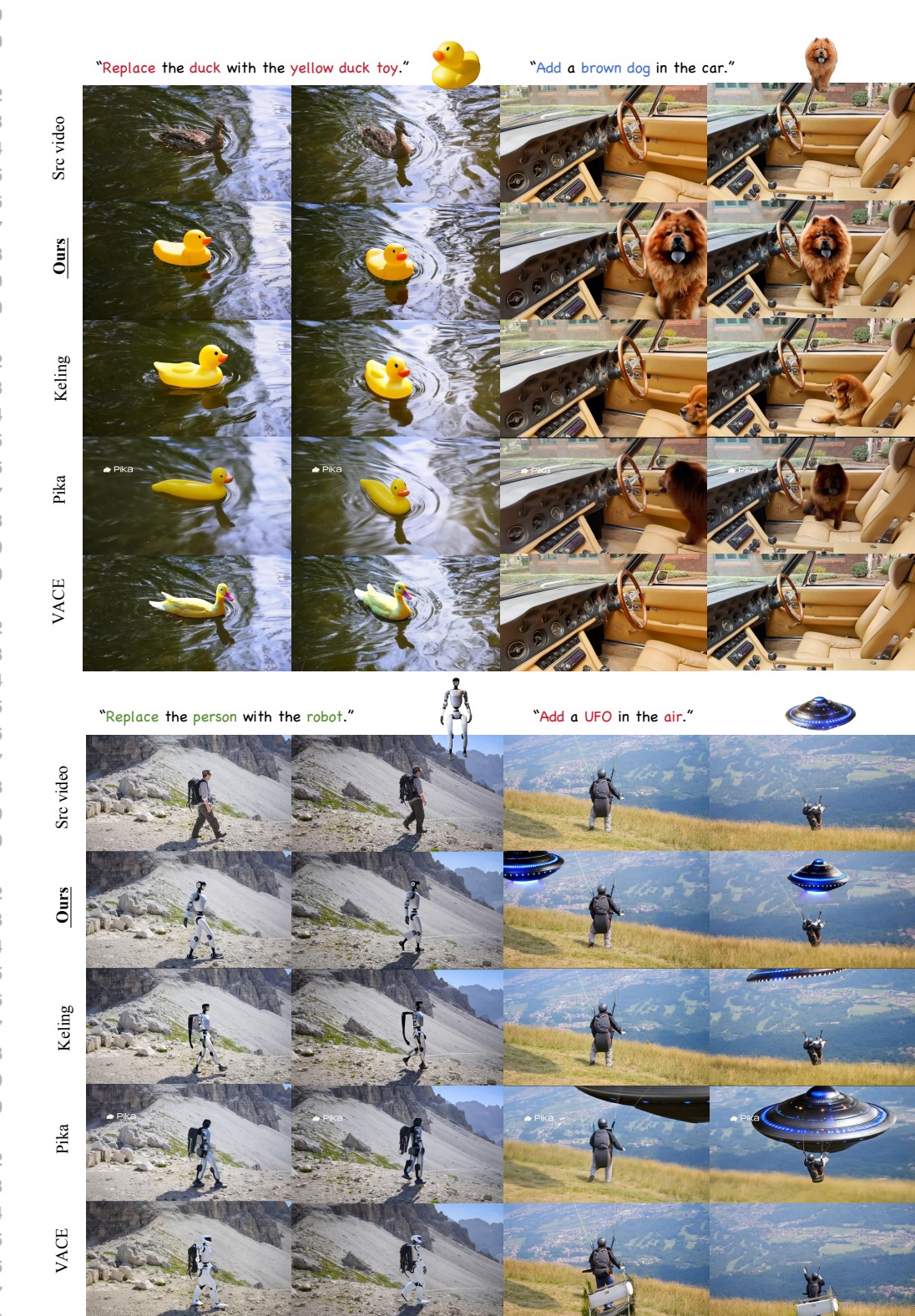

Figure 22: Visual comparsion on VIE-Bench.

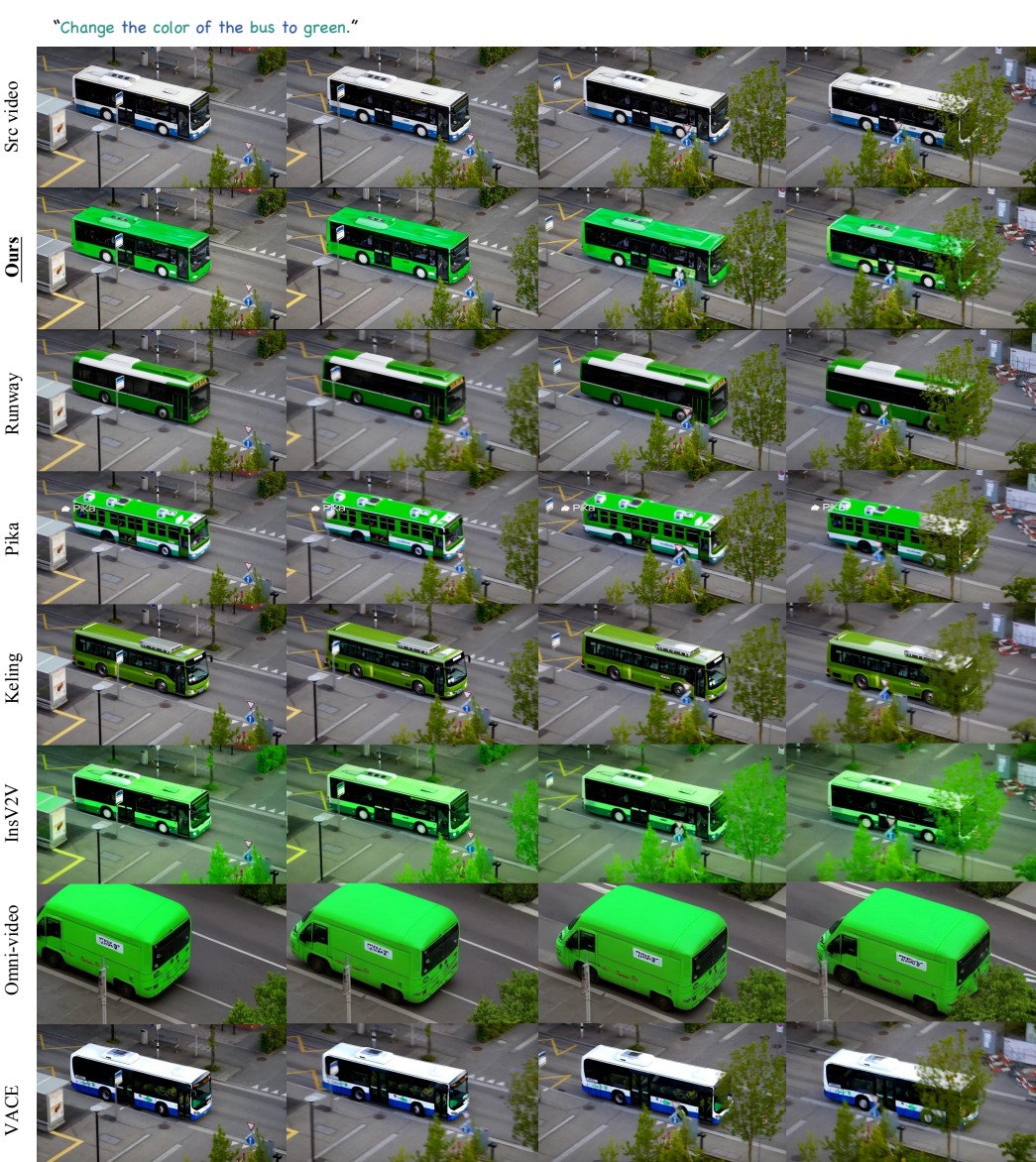

Figure 23: Visual comparsion on VIE-Bench.

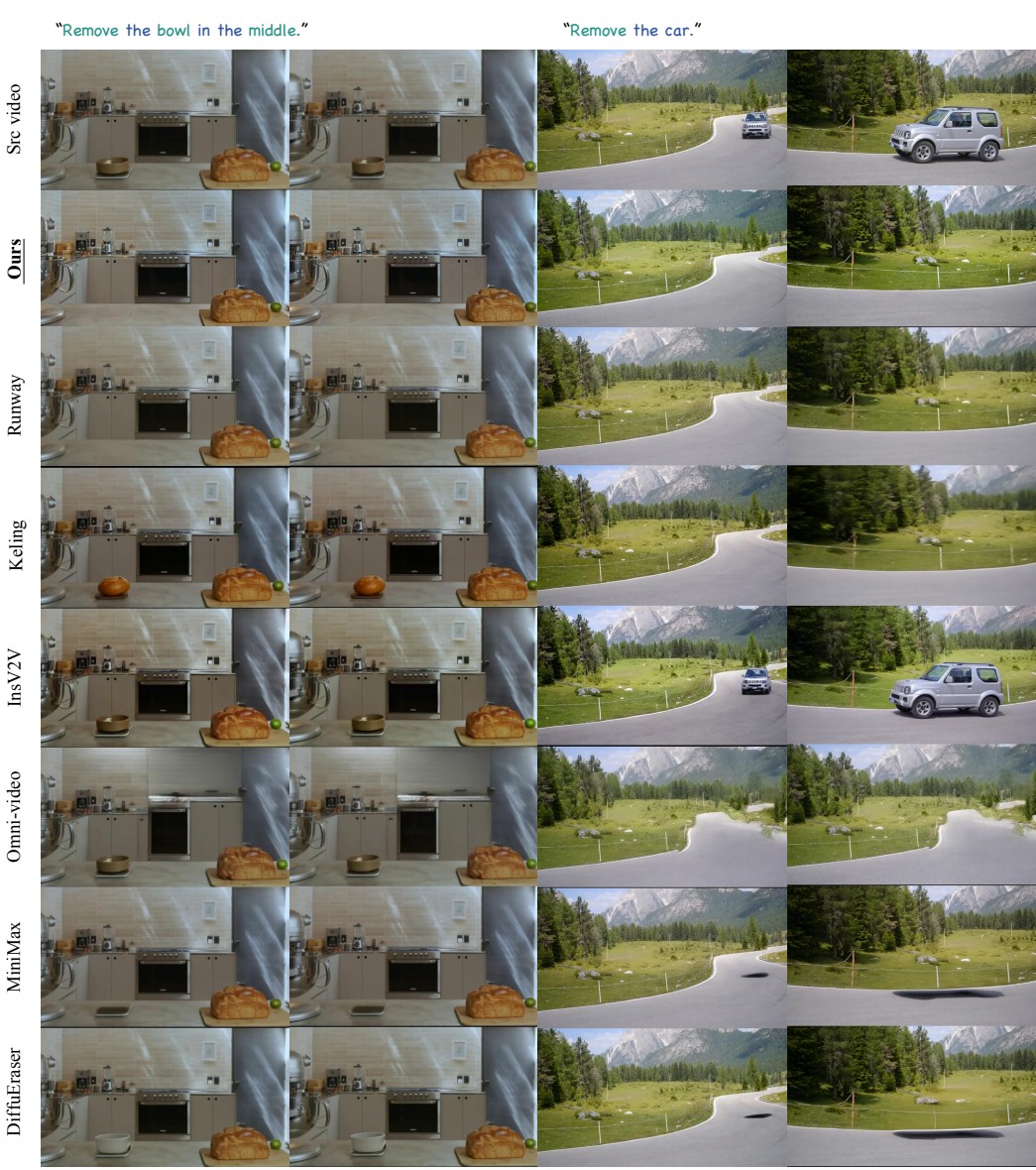

Figure 24: Visual comparsion on VIE-Bench.

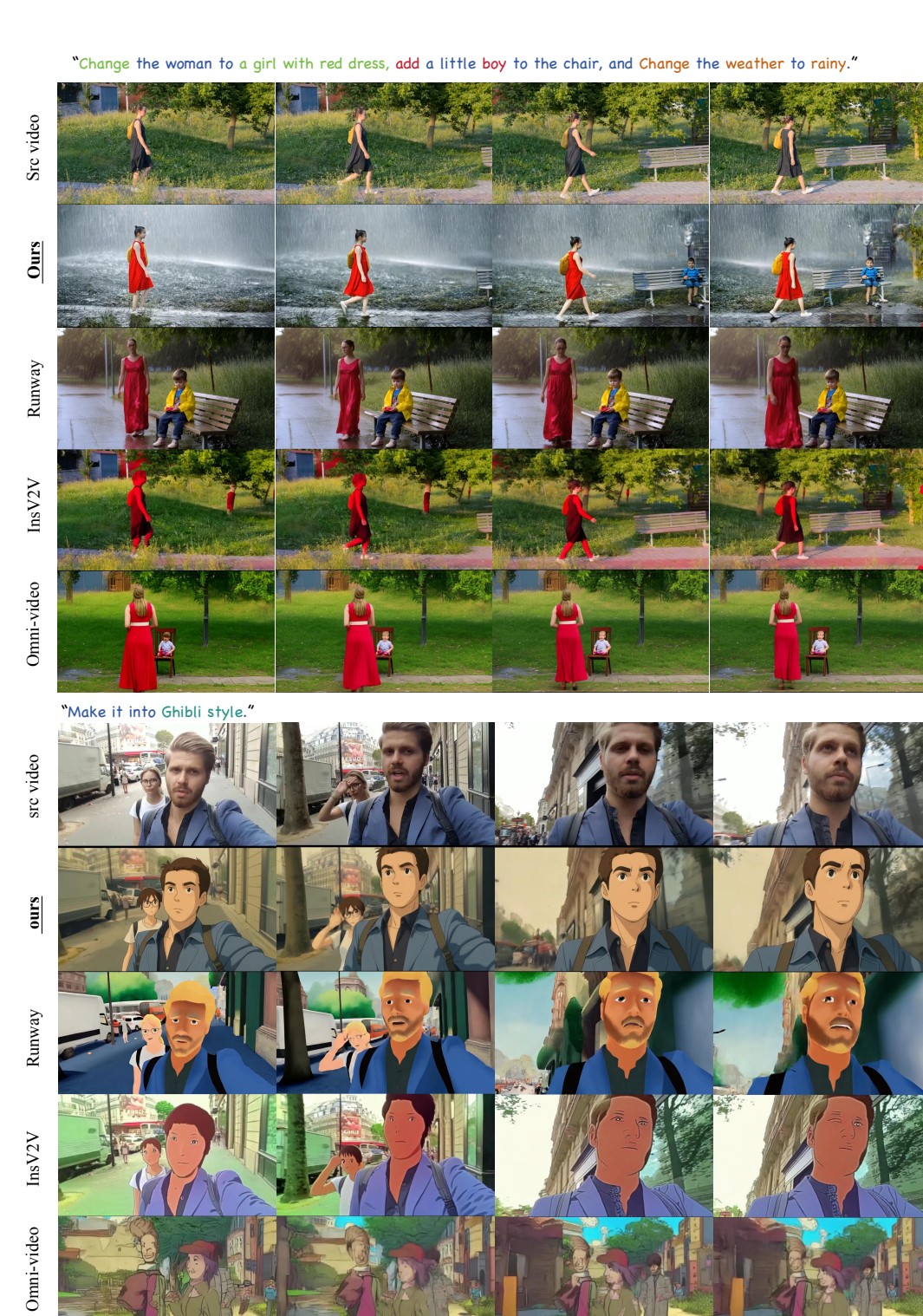

Figure 25: Visual comparsion on VIE-Bench.

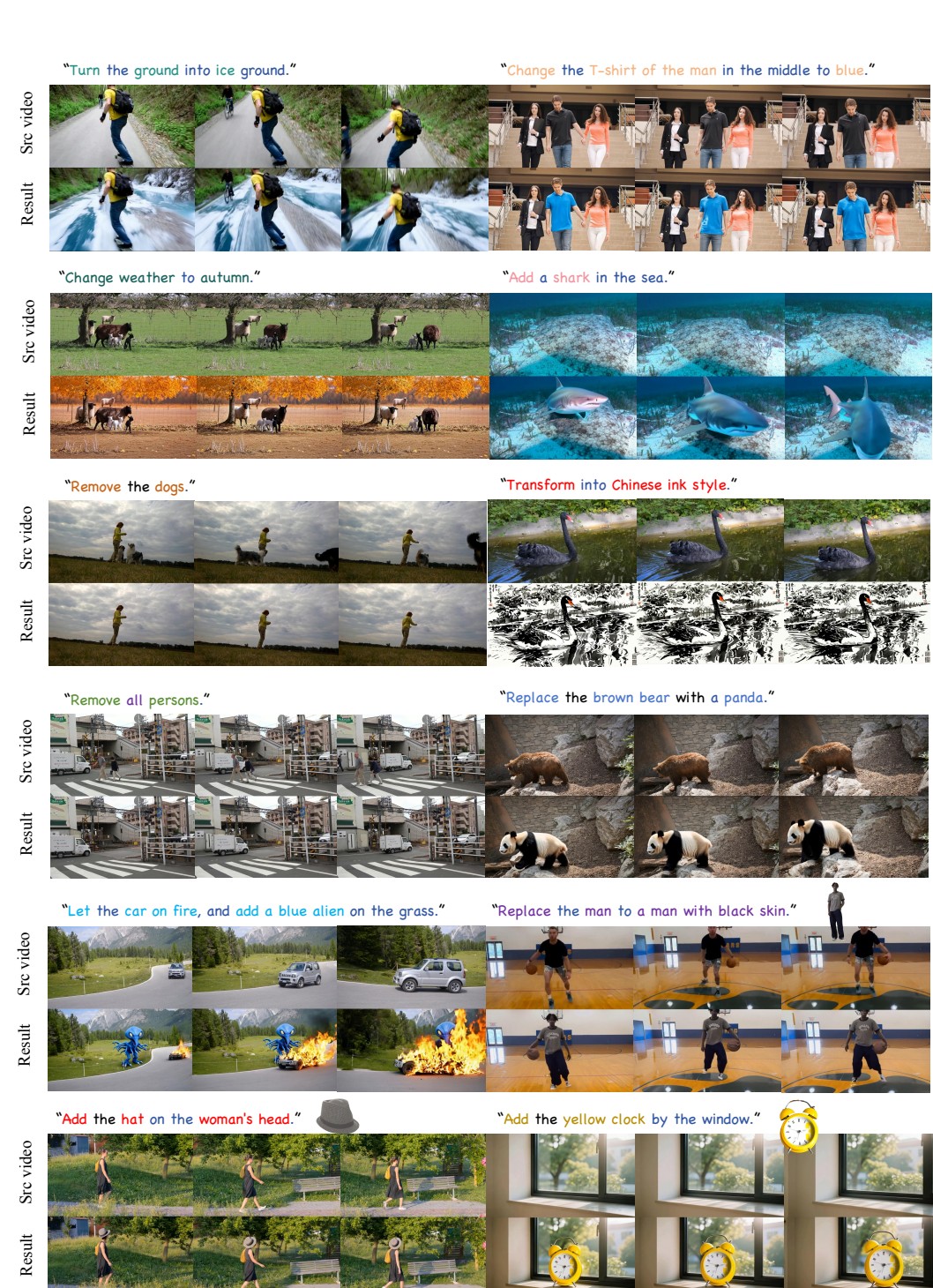

Figure 26: More video editing results of our method.

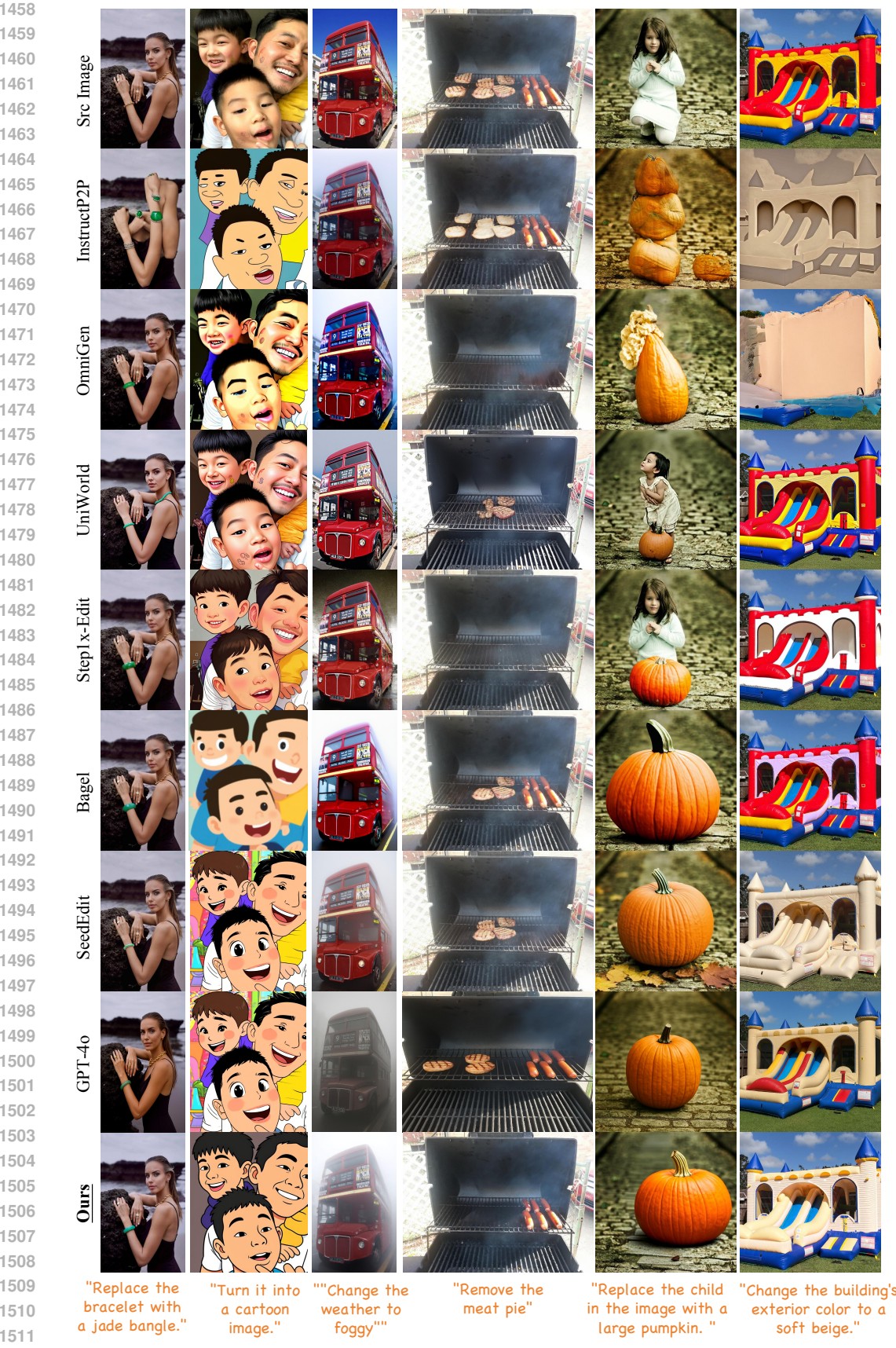

Figure 27: Visual comparsion on image editing.

