# OpenReview forum: "InstructX: Towards Unified Visual Editing with MLLM Guidance"
_ICLR.cc/2026/Conference — Submitted to ICLR 2026_

### Official Review · Reviewer_6Kao · 2025-10-30

**Soundness:** 3
**Presentation:** 3
**Contribution:** 3
**Rating:** 6
**Confidence:** 4

**Summary:**

This paper propose InstructX which integrates MLLMs with diffusion models to perform instruction-driven editing tasks across both images and videos within a single model and it explores optimal design choices for combining MLLMs and diffusion models, emphasizing that MLLMs should actively participate in the editing process rather than being treated as mere feature extractors.

The framework leverages large-scale image editing data to train video editing capabilities, addressing the scarcity of high-quality video editing datasets.

 The paper introduces VIE-Bench, a high-quality benchmark for instruction-based video editing, comprising diverse tasks such as object addition, removal, style change, and reference-based edits.

Extensive experiments demonstrate that InstructX outperforms open-source methods and competes with closed-source solutions in both image and video editing tasks.

**Strengths:**

It critically evaluates different architectural approaches for combining MLLMs and diffusion models (fig3), offering evidence-based recommendations for optimal integration. ​

It proposes a novel approach to address the lack of high-quality video editing datasets by leveraging image data to train video editing models. ​

The introduction of VIE-Bench provides a new standard for evaluating instruction-based video editing methods, which can be used for future research comparisons. ​

**Weaknesses:**

The main weakness of this paper is that its overall takeaway message feels rather weak. While the architectural design presented is a valid and potentially useful contribution from which readers can learn, it is unclear what additional insights or lessons the audience can gain beyond this. The paper reads more like a technical report than a research study. I encourage the authors to provide deeper insights and discussions that can inspire readers, rather than merely presenting results

**Questions:**

na

---

> ### Author Response · Authors · 2025-11-21
> **Responses to Review Comments**
>
> **`6Kao-Q1`: The main weakness of this paper is that its overall takeaway message feels rather weak**
>
> We sincerely thank the reviewer for the constructive feedback.
> In the paper, we aim to answer three fundamental research questions regarding the integration of MLLMs and diffusion models for editing:
> - Q1: Can a large amount of high-quality image-editing data be used to improve video editing tasks?
> - Q2: Do MLLMs actually help with visual editing tasks, and by how much specifically?
> - Q3: Among the many ways to combine MLLMs and diffusion models, which is better suited for editing tasks?
>
> Our paper provides evidence-based answers to these questions, which constitute our core contributions/takeaways:
> - **Insight 1 (Data Transferability): Yes.** As demonstrated in Figure 6, training on high-quality image editing data enables significant capability transfer to video editing. This unlocks powerful zero-shot abilities on video tasks, addressing the scarcity of high-quality video editing datasets.
> - **Insight 2 (MLLM Necessity):** The MLLM is critical. As shown in Figures 14-15, our MLLM-integrated model consistently outperforms diffusion-only baselines across all tasks. This proves that MLLMs contribute essential semantic understanding and instruction-following capabilities that diffusion models alone may lack.
> - **Insight 3 (Optimal Architecture):** As shown in Figure 4, among the four structures we validated, the combination of **Meta-query + LoRA fine-tuned MLLM + Small Connector** achieves the best performance. The key insight here are that the editing should be accomplished within the MLLM itself; LoRA is helpful for adapting the MLLM to the editing task; and a large connector is unnecessary.
>
> These insights cannot be derived from previous work, nor have they been systematically validated in prior studies. We believe that these insights are important for the future development of the editing field.
>
> We will revise the manuscript to explicitly highlight these three insights and the "Why" behind our architectural choices to strengthen the overall takeaway message.

---

> > ### Author Response · Authors · 2025-11-29
> >
> > Dear reviewer, the deadline for the discussion is very close, and we would like to know if you have any questions or concerns.

---

### Official Review · Reviewer_PmDh · 2025-10-31

**Soundness:** 3
**Presentation:** 3
**Contribution:** 2
**Rating:** 4
**Confidence:** 5

**Summary:**

This paper presents InstructX, a unified framework for instruction-guided image and video editing, which leveraging multimodal large language models (MLLMs) to improve diffusion-based editing quality. The core idea is to use a MLLM to extract related text prompt feature to replace the original text input of the diffusion model. The MLLM processes the instruction and source visual content, producing learning meta-queries that guide a DiT to perform the edit. During training, three training stages are performed, including feature alignment, joint training, and quality fine-tuning. The author claimed that training on image data can emerge video editing capabilities without explicit supervision.

**Strengths:**

1.	A unified design that seamlessly handles both image and video editing in a single model.

2.	Introduces VIE-bench, a high-quality, instruction-based video editing benchmark with 140 examples across 8 categories.

3.	The method outperforms many open-source methods and competes with closed-source models.

**Weaknesses:**

1.	My biggest concern is the novelty of the proposed method. I think the core idea of InstructX is very similar to metaQuery, including the core technique used. Compared to metaQuery, the differences lie in 1) InstructX uses Lora in MLLM while metaQuery simple freezes the MLLM, 2) InstructX uses lightweight MLP as the connector, while metaQuery use a relatively large transformer. There two modifications are good, and make the model well, but I don’t think the novelty is sufficient.

2.	While the author said the training on image data can emerge video editing capabilities, the method still needs video editing training data. Is it possible to completely get rid of video data?

**Questions:**

Please refer to the weaknesses.

---

> ### Author Response · Authors · 2025-11-21
> **Responses to Review Comments**
>
> **`PmDh-Q1`: The core idea of InstructX is very similar to MetaQuery, thus the novelty is limited.**
>
> We respectfully wish to clarify the scope and contributions of our work. Our primary goal is not merely to optimize the MetaQuery architecture, but to conduct a systematic investigation into **how to optimally integrate MLLMs with Diffusion models** for instruction-guided editing.
> To this end, we performed extensive ablation studies to identify the most effective integration paradigm (please refer to the detailed insights in our response to `6Kao-Q1`). Our final architecture—utilizing LoRA fine-tuning and a lightweight connector—is the result of this rigorous empirical analysis, which proves that a massive connector is unnecessary and that the editing reasoning should reside within the MLLM.
> Furthermore, we highlight two critical contributions beyond the architecture:
> - **Data-Centric Discovery**: We demonstrate that training on high-quality image data can effectively "unlock" zero-shot video editing capabilities for tasks unseen during training. This is a significant finding that addresses the scarcity of high-quality video editing datasets.
> - **Efficiency & Accessibility**: InstructX achieves SOTA performance—comparable to closed-source commercial models—while converging rapidly on standard compute resources (32-64 GPUs). We believe this balance of high performance and low computational cost provides a valuable reference for the community, particularly for research institutions with limited resources.
>
> ---
> **`PmDh-Q2`: Is it possible to completely get rid of video data?**
>
> From a technical perspective, completely eliminating video data remains a challenge. Since the backbone is a video generation model, fine-tuning exclusively on image data typically leads to **catastrophic forgetting of temporal dynamics**, causing the model to lose its ability to generate coherent videos. We add a discussion in Fig.15 in Sec.A.4 (marked in green). The results demonstrate that using only image data disrupts the temporal consistency of video generation outcomes, leading to undesired flickering and artifacts.
>
> Therefore, our objective is not to discard video data entirely, but to **leverage abundant high-quality image data to augment video editing capabilities**. This strategy effectively alleviates the bottleneck caused by the current scarcity of high-quality video editing datasets. While we anticipate more mature video data collection pipelines in the future, our approach offers a highly effective solution to the data shortage problem in the current landscape.

---

> > ### Author Response · Authors · 2025-11-29
> >
> > Dear reviewer, the deadline for the discussion is very close, and we would like to know if you have any questions or concerns.

---

### Official Review · Reviewer_jWAu · 2025-11-01

**Soundness:** 2
**Presentation:** 3
**Contribution:** 2
**Rating:** 4
**Confidence:** 5

**Summary:**

InstructX presents an end-to-end framework that unifies image and video instruction editing. A LoRA-tuned Qwen-VL-3B serves as the understanding module; 256 learnable queries are appended for images and 512 for videos, and only the hidden states of these queries are retained as “editing features.” A lightweight two-layer MLP replaces the large Transformer connector used in prior MetaQuery-style systems, mapping the query features to the text channel of a Wan 2.1 DiT decoder. Training is carried out in three stages: (1) image-only feature alignment, (2) mixed image–video training that injects the source VAE latent into the noise to boost fidelity and elicits zero-shot video capability, and (3) a small high-quality finetuning stage. Evaluated on ImgEdit-Bench and GEdit-Bench for images and the authors’ VIE-Bench for videos, InstructX attains better overall scores among open-source methods.

**Strengths:**

1.	The paper is well structured and easy to follow.

2.	The “small Query + LoRA + lightweight MLP” design matches the performance of large Transformer connectors while lowering memory and compute cost.

3.	The evaluation is comprehensive, spanning multiple public benchmarks for both image and video editing.

**Weaknesses:**

1.	Limited novelty: the approach mainly extends the MetaQuery “query + connector” paradigm, replacing the large Transformer with a small MLP and adding extra VLM fine-tuning; the claimed unified image/video capability mainly relies on the pre-trained Wan 2.1 backbone rather than a new algorithmic contribution.

2.	The video branch employs a fixed set of 512 queries; whether this is sufficient for long or highly complex videos—and thus able to capture long-range temporal semantics—remains unverified.

3.	The paper lacks experiments or discussion on high-resolution video editing, leaving scalability to larger resolutions unexplored.

**Questions:**

N/A

---

> ### Author Response · Authors · 2025-11-21
> **Responses to Review Comments**
>
> **`jWAu-Q1`: Limited novelty**
>
> We respectfully wish to clarify the scope and contributions of our work. Our primary goal is not merely to optimize the MetaQuery architecture, but to conduct a systematic investigation into **how to optimally integrate MLLMs with Diffusion models** for instruction-guided editing.
> To this end, we performed extensive ablation studies to identify the most effective integration paradigm (please refer to the detailed insights in our response to `6Kao-Q1`). Our final architecture—utilizing LoRA fine-tuning and a lightweight connector—is the result of this rigorous empirical analysis, which proves that a massive connector is unnecessary and that the editing reasoning should reside within the MLLM.
> Furthermore, we highlight two critical contributions beyond the architecture:
> - **Data-Centric Discovery**: We demonstrate that training on high-quality image data can effectively "unlock" zero-shot video editing capabilities for tasks unseen during training. This is a significant finding that addresses the scarcity of high-quality video editing datasets.
> - **Efficiency & Accessibility**: InstructX achieves SOTA performance—comparable to closed-source commercial models—while converging rapidly on standard compute resources (32-64 GPUs). We believe this balance of high performance and low computational cost provides a valuable reference for the community, particularly for research institutions with limited resources.
>
> ---
>
> **`jWAu-Q2`: Ablation on video queries.**
>
> The table below presents a comparison of different video-query lengths on the VIE benchmark, and the overall differences are small. Increasing the video-query length does not lead to significant performance gains. This is primarily because the VLM mainly provides high-level semantic information. We add this discussion in Sec.A.4 (marked in red) in the paper.
>
> |               | \|-------- | 5,000 iter | --------\| | \|--------  | 10,000 iter | --------\| |
> | ------------- | --------- | --------- | --------- | --------- | -------- | -------- |
> |               | Instruct follow | Preservation    | Quality    | Instruct follow | Preservation   | Quality   |
> | 512 video query | 8.61     | 8.82     | 7.94     | 8.55     | 8.81    | 7.90    |
> | 1024 video query | 8.85     | 8.98     | 8.10     | 8.70     | 8.91    | 8.02    |
>
> ---
>
> **`jWAu-Q3`: The paper lacks experiments or discussion on high-resolution video editing, leaving scalability to larger resolutions unexplored.**
>
> We need to clarify that in the **Limitations** section, we explain that our method is limited in high-resolution editing tasks. The restriction primarily stems from the underlying video generation backbone, which is not natively designed for high-resolution synthesis, combined with the substantial computational costs required for high-res training.
>
> Nevertheless, we observe that InstructX exhibits strong **resolution generalization**. Even when trained on lower-resolution datasets, it effectively handles high-resolution (1080P) inference. We add a discussion in Sec.A.6 (marked in red), showing the promising result in 1080P videos. While there is a **slight performance gap** compared to the low-resolution setting, this is **consistent with our expectations** given the backbone and training resources constraints discussed above.

---

> > ### Author Response · Authors · 2025-11-29
> >
> > Dear reviewer, the deadline for the discussion is very close, and we would like to know if you have any questions or concerns.

---

### Official Review · Reviewer_7XS6 · 2025-11-01

**Soundness:** 3
**Presentation:** 3
**Contribution:** 3
**Rating:** 6
**Confidence:** 4

**Summary:**

The authors propose InstructX, a unified framework for image and video editing that integrates MLLM with diffusion models. They adopt Separate Learnable Queries for image and video, finetune the MLLM using LoRA, and introduce a lightweight MLP connector to effectively inject the understanding capabilities of MLLM into the diffusion model. Extensive experiments on image and video editing tasks demonstrate the superior performance of InstructX. Additionally, the authors introduce VIE-Bench, a benchmark specifically designed to evaluate instruction-based video editing capabilities.

**Strengths:**

1. Well-designed architecture. InstructX leverages Learnable Queries (similar to MetaQuery) to enable effective modality interaction. Rather than treating MLLM as feature extractors that freeze the MLLM and train a large connector, the framework finetunes the MLLM with LoRA and uses a lightweight MLP connector, enabling faster convergence and improved performance.

2. Well-designed data curation and training strategy. The authors create a large-scale training dataset, and their training strategy is well-motivated.
3. Comprehensive evaluation. Comprehensive experiments on ImgEdit-Bench, GEdit-Bench, and proposed VIE-Bench show high performance on instruction-based image and video editing tasks.

**Weaknesses:**

Interestingly, InstructX exhibits video segmentation and style transfer abilities, which are absent from the video training data but present in the image data. It would strengthen the paper to provide quantitative results comparing model performance when trained with only video data versus both image and video data, particularly on segmentation and style transfer tasks.

**Questions:**

1. How to balance the training of image data and video data or different stages? Any ablations?
2. Where can we find the model size of this method, and how can we compare it with other methods?
3. How to handle the longer video editing in this framework?

---

> ### Author Response · Authors · 2025-11-21
> **Responses to Review Comments**
>
> **`7XS6-Q1`: Please compare training only with videos and training with mixed image–video data on the tasks of video segmentation and video stylization.**
>
> Thanks for this suggestion. We add ablation experiments on the role of image data in Sec.A4 (marked in blue). As illustrated in Fig.14, the model trained exclusively on video data fails to respond to instructions for segmentation and stylization. This is primarily because current video editing datasets lack the specific tasks.
>
> In contrast, mixed training successfully unlocks these capabilities. By incorporating high-quality image editing data (which contains image stylization segmentation samples), the model learns to transfer these semantic editing abilities to the video domain, enabling effective video segmentation and stylization.
>
> ---
>
> **`7XS6-Q2`: How to balance the training of image data and video data? Any ablations?**
>
> The table below compares the effects of different image–video ratios on the VIE benchmark. It can be observed that the model's performance is not highly sensitive to this ratio. Nevertheless, image data remains essential for broadening the scope of video editing applications, while video data is necessary to preserve the model's temporal generation capability. We update the table in Tab.5 in the main paper.
>
> |               | \|-------- | 5,000 iter | --------\| | \|--------  | 10,000 iter | --------\| |
> | ------------- | --------- | --------- | --------- | --------- | -------- | -------- |
> |    image:video           | Instruct follow | Preservation    | Quality    | Instruct follow | Preservation   | Quality   |
> | 2:3 (paper setting)  | 8.40     | 8.73     | 7.77     | 8.26     | 8.73    | 7.59    |
> | 1:4 | 8.36     | 8.74     | 7.67     | 7.91     | 8.54    | 7.38    |
> | 4:1 | 8.41     | 8.79     | 7.74     | 8.69     | 8.92    | 7.87    |
>
> ---
>
> **`7XS6-Q3`: Where can we find the model size of this method, and how can we compare it with other methods?**
>
> Thanks for this helpful suggestion. In the table below, we present the model parameters of representative image editing and video editing methods. It can be seen that the number of model parameters in our method is comparable to that of mainstream approaches. The comparaison in the paper(Tab.1,2,3) demonstrate that our performance surpasses these methods. We update this model size comparison in Sec.A.7 (marked in blue) in this paper.
>
> |             | model size               |
> |-------------|--------------------------|
> | OmniGen     | 3.8B                     |
> | Step1X-Edit | 12.5B                    |
> | UniWorld    | 12B                      |
> | Bagel       | 14.6B                    |
> | Omni-Video  | 11B                      |
> | VACE        | 14B                      |
> | ours        | 14B(DiT)+0.6B(MLLM LoRA) |
>
> ---
>
> **`7XS6-Q4`: How to handle the longer video editing in this framework?**
>
> Thanks for this helpful question. Inspired by your question, we modify the inference pipeline to enable long video editing. Specifically, we process long videos using a sliding window approach, where consecutive windows overlap at the tail frame of the previous window and the head frame of the next window. During editing, the editing result of the tail frame of the preceding window replaces the head frame of the subsequent window to maintain consistency between windows. We include a visualization of this process in Fig.16 in Sec.A.5 (marked in blue). In our tests, using a 5-second window, we successfully edit a 30-second video at 30 FPS. The results show smooth transitions between windows, demonstrating that our method can be effectively extended to long video editing.

---

> > ### Author Response · Authors · 2025-11-29
> >
> > Dear reviewer, the deadline for the discussion is very close, and we would like to know if you have any questions or concerns.

---

### Author Response · Authors · 2025-11-21
**Common Response**

We thank all the reviewers for their efforts in improving our paper to a higher standard. We have addressed all your concerns and update the manuscript accordingly. If any further questions, please let us know—we welcome any discussion.

---

### Meta-Review · Area_Chair_EiKL · 2026-01-06

**Summary:**

This paper proposes InstructX, a unified framework for image and video editing, and VIE-Bench for evaluating instruction-based video editing. Experiments demonstrate that high-quality image editing data can boost the zero-shot performance of corresponding video editing tasks. Given the scarcity of large-scale, high-quality video datasets, this finding provides valuable guidance for future studies in video editing.

**Reviewer Concerns:**

Reviewers generally acknowledged the well-designed architecture, the introduction of VIE-Bench, and the comprehensive evaluation. Key concerns concentrate on the novelty, with critics noting the similarity to MetaQuery.

**Reviewer Scores:**

The paper initially received mixed reviews, with 2 marginally below the acceptance threshold and 2 marginally above the acceptance threshold. In rebuttal, the authors clarified their contributions lie in systematic investigation of MLLM-diffusion integration. However, since the investigated architectures are similar, the AC agrees with Reviewer 6Kao that the investigation is more like a technical report rather than a research. As for the claim that image editing data boosts zero-shot video editing tasks, Reviewer 7XS6 requested for quantitative results when image data is completely excluded. However, the authors only provided qualitative results in the revised manuscript, leaving the request partially unresolved. After considering all the reviews and rebuttals, the AC recommends rejecting the paper.

---

### Decision · Program_Chairs · 2026-01-26

Reject